# Pupil size reveals arousal level fluctuations in human sleep

Manuel Carro-Domínguez [1], Stephanie Huwiler [1], Stella Oberlin[1], Timona Leandra Oesch [1], Gabriela Badii[2], Anita Lüthi [3], Nicole Wenderoth [1,4,5,7], Sarah Nadine Meissner[1,7] & Caroline Lustenberger [1,5,6,7] ✉

Recent animal research has revealed the intricate dynamics of arousal levels that are important for maintaining proper sleep resilience and memory consolidation. In humans, changes in arousal level are believed to be a determining characteristic of healthy and pathological sleep but tracking arousal level fluctuations has been methodologically challenging. Here we measured pupil size, an established indicator of arousal levels, by safely taping the right eye open during overnight sleep and tested whether pupil size affects cortical response to auditory stimulation. We show that pupil size dynamics change as a function of important sleep events across different temporal scales. In particular, our results show pupil size to be inversely related to the occurrence of sleep spindle clusters, a marker of sleep resilience. Additionally, we found pupil size prior to auditory stimulation to influence the evoked response, most notably in delta power, a marker of several restorative and regenerative functions of sleep. Recording pupil size dynamics provides insights into the interplay between arousal levels and sleep oscillations.

Arousal level can be viewed as a continuum of activation of cortical and subcortical networks that influences how stimuli are processed across all vigilance states. During wakefulness, arousal levels are closely tied to alertness, while during sleep, according to rodent studies, drivers of arousal levels are strongly suppressed during non-rapid eye movement (NREM) and rapid eye movement (REM) sleep[1,2]. Central drivers of arousal levels include brain areas such as the noradrenergic (NA) locus coeruleus (LC)[3], the cholinergic nucleus basalis of Meynert[4], serotonergic neurons in the dorsal raphae[5], and orexin/hypocretin-producing neurons in the lateral hypothalamus[6]. Activity changes in these arousal level-regulating brain areas have further been linked to pupil size changes, meaning changes in pupil size can index their activity[7–10]. Other drivers of

arousal levels that project to these nuclei include neuropeptide S[11], the cortex[12], and the autonomic nervous system[13]. Arousal level dynamics can be large and sustained, resulting in transitions from low vigilance states of sleep, such as NREM sleep, to larger states manifesting as wakefulness, or from low to even lower states manifesting as REM sleep. Alternatively, they can be subtle and temporary, detected primarily by neurophysiological markers such as K-complexes[14,15], spindles[16], or sleep arousals, defined as very short bouts of wakefulness manifesting as abrupt but temporary changes in polysomnographic (PSG) signals[17]. However, little is known about whether similar arousal level dynamics exist also during human sleep and how they relate to sleep macrostructure and microstructural events. To answer this question, we measured

[1]Neural Control of Movement Lab, Institute of Human Movement Sciences and Sport, Department of Health Sciences and Technology, ETH Zurich, 8092 Zurich, Switzerland. [2]FMH Ophthalmology, Zurich, Switzerland. [3]Department of Fundamental Neurosciences, University of Lausanne, Lausanne, Switzerland. [4]Future Health Technologies, Singapore-ETH Center, Campus for Research Excellence and Technological Enterprise (CREATE), Singapore, Singapore. [5]Neuroscience Center Zurich (ZNZ), University of Zurich, ETH Zurich, Zurich, Switzerland. [6]Center of Competence Sleep & Health Zurich, University of Zurich, Zurich, Switzerland. [7]These authors contributed equally: Nicole Wenderoth, Sarah Nadine Meissner, Caroline Lustenberger. ✉e-mail: caroline.lustenberger@hest.ethz.ch

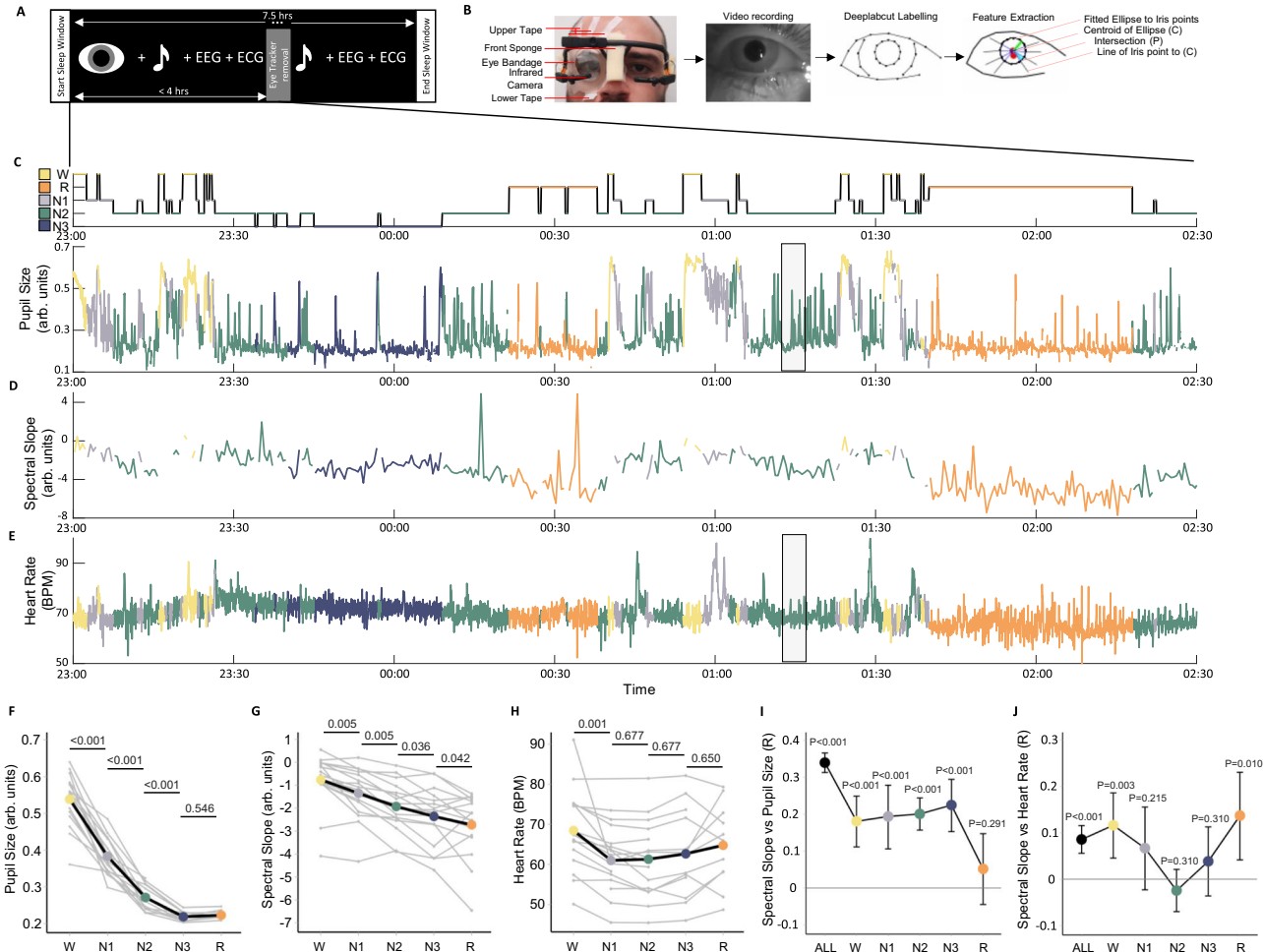

**Fig. 1 | Experimental procedures. A** Overall sleep protocol: Auditory stimulation was applied during non-rapid eye movement (NREM) sleep, whereas pupil size was measured only for up to 4 h. **B** Pupillometry protocol: The eye was taped open and protected with a transparent bandage while tracked with a camera. Eye features were extracted from the videos: pupil and eyelid centroids (red markers), fitted ellipse around pupil circumference (black ellipse), major and minor axis of pupil ellipse (cyan and green line), iris points connected to pupil centroid (center black lines). **C–J** W: Wakefulness (yellow); N1: NREM1 (gray); N2: NREM2 (green); N3: NREM3 (blue); R: REM (orange). Consent was obtained for the publication of this image. **C–E** Hypnogram while pupil size was recorded and accompanying time series from a representative participant. The displayed time period refers to the first 3.5 h of a recording, as shown in (**A**). **C** Pupil size fluctuations during sleep. **D** 30-s segments of the spectral slope without artifacts. **E** Heart rate (HR) in beats per minute (BPM). The gray box is the time period plotted in the bottom plot of

Fig. 2A. **F** Average pupil size ($F_{(4, 57.43)} = 168.90$, $p < 0.001$), **G** spectral slope ($F_{(4, 68)} = 41.37$, $p < 0.001$), and **H** HR ($F_{(4, 55.52)} = 7.623$, $p < 0.001$) for wake, N1, N2, N3, and REM sleep. The data of each participant is depicted in gray. Statistical differences between sleep stages are depicted (see supplementary Tables 1–3 for more detailed information). **I** Repeated measures correlation within sleep stages where pupil data was aggregated in 30-s segments to match the segment lengths of the spectral slope data. "All" refers to all sleep stages together (All: $n = 17$, W: $n = 16$, N1: $n = 17$, N2: $n = 17$, N3: $n = 3$, R: $n = 12$). **J** Repeated measures correlation within sleep stages where HR was aggregated in 30 s segments to match the segment lengths of the spectral slope data (All: $n = 17$, W: $n = 16$, N1: $n = 17$, N2: $n = 17$, N3: $n = 3$, R: $n = 12$). 30-s segments where auditory stimulation had been applied were excluded. $p$-values are based on a post hoc $t$-test and are adjusted for multiple comparisons using Benjamini-Hochberg correction. Data are presented as mean values +/− 95% confidence intervals. All statistical results are based on two-sided tests.

pupil size to index arousal levels (Fig. 1), together with polysomnography (PSG) and electrocardiography (ECG).

The locus coeruleus (LC), a small brainstem nucleus and the main source of noradrenergic transmission (NA), has been recently discovered in rodents to be causally mediating infraslow fluctuations (ISFs) in arousal levels lasting 30–60 s during NREM sleep that represent vigilant and consolidated periods of sleep[1,2]. In mice, ISFs in NA-driven arousal levels are linked to the thalamocortico-cortical activity in the forebrain. This activity includes the emergence of sleep spindles (transient bursts of sigma oscillations in the 12-16 Hz range) that are regulated by NA-dependent control of the membrane potential of thalamic neurons[2,16,18]. NA levels show an anti-correlation with spindle activity in the cortex and a positive correlation with heart rate (HR). Specifically, consolidated periods of low arousal levels associated with low LC-NA activity coincide with an increased likelihood of sleep

spindle clusters, which can protect sleep from external disturbances, and a decrease in HR[2,16,18,19]. Vigilant periods of high arousal levels associated with high LC-NA activity, by contrast, coincide with low sleep spindle probability and elevated HR. Such periods of elevated arousal levels during NREM sleep might be essential for survival in mice due to the omnipresent environmental threats. Furthermore, recent studies in rodent models indicate that the role of arousal level dynamics during sleep goes beyond changes in vigilance and contributes fundamentally to the macro- and micro-architecture of sleep, coordination of the brain and autonomic system, and memory consolidation, potentially playing a fundamental role in restorative processes of sleep (for review, see ref. 2).

Yet, it is unclear whether ISFs of arousal levels are preserved in human sleep. Initial evidence indicates that sleep spindles in NREM sleep exhibit an ISF in humans similar to the rodent model[20,21].

However, establishing a systematic link between these spindle fluctuations and changes in arousal levels, most likely driven by the LC, has been challenging due to the lack of appropriate methods for estimating LC activity during human sleep. Here, we propose the use of pupillometry as an approach to test several predictions from the rodent literature in human sleep. It is well-established that the size of the eye's pupil can serve as an indirect marker of arousal levels when light conditions are kept constant[22]. Even though several neuromodulatory systems have been associated with these non-luminance-related changes in pupil size, the LC-NA system has so far received the strongest evidence[7,8,23–27]. Furthermore, pupil size and cortical activity have been measured simultaneously during sleep in mice, revealing ISFs of pupil size during NREM sleep that are anti-correlated with electrocortical activity in the spindle range[19]. This finding aligns with previous studies showing LC-NA activity fluctuating with spindle ISFs and indirectly suggests that pupillometry is a valuable tool for indexing arousal levels during sleep and providing a proxy of LC-NA activity during NREM sleep[2,16,18,25,28].

Even though pupillometry is a standard research tool in human participants during wakefulness, very little data has been published reporting human pupil dynamics during sleep[29]. We have, therefore, developed a safe method for measuring changes in pupil size during sleep using a conventional, low-dose infrared system. Here we synchronized our approach with PSG and ECG. We were able to demonstrate that arousal levels fluctuate in human sleep, as indexed by pupil size dynamics across and within sleep stages. We further established that these fluctuations surround markers of cortical arousal levels, sleep electroencephalography (EEG) micro-architectures such as spindles and K-complexes, and HR. Using auditory stimulation during NREM sleep, we further demonstrate that the brain response to administered acoustical stimuli changes depending on arousal levels indexed by differences in pupil size.

## Results

We developed a method for measuring pupil size together with PSG and ECG in healthy participants ($n = 17$) during sleep (Fig. 1A, B and Supplementary Movie 1). Portable pupillometry via a low-dose infrared system was applied to the right eye, which was kept open by applying three strips of tape on the upper eyelid and one wider strip of tape on the lower eyelid. To reduce discomfort, the eye was taped open without exceeding the participant's palpebral fissure during wakefulness. To prevent eye dryness, vitamin A-containing eye ointment was applied to the eye immediately before the eye was covered by a transparent eye bandage. Eye bandages are routinely used in nocturnal lagophthalmos patients who cannot close their eyelids during sleep to protect the eye from dust and mechanical damage[30]. All participants could tolerate having their eyes taped open and were able to fall asleep. While there were changes in sleep macrostructure between taping the right eye open in the first half of the night and having it closed in the second half of the night, these changes were consistent with a typical night's sleep, and no major disturbances were found from sleeping with the eye open (see Supplementary Fig. 1A for sleep macrostructural difference when the right eye is open versus closed). However, to truly answer this question, one would have to run a larger study with multiple sleep sessions in the laboratory. Data acquisition started once the room was completely dark, except for the infrared light source emitted by the pupillometer, ensuring all vigilance states had exactly the same lighting conditions. In the debriefing, 2 out of 17 participants reported waking up due to the unfamiliarity of sleeping with one eye open. Participants slept in the lab for 7.5 h with sleep onset having been adjusted to each individual's typical bedtime (around 10–11 pm). Auditory stimulation was applied during NREM sleep throughout the night. PSG and ECG were acquired during the whole sleep session, but pupillometry was limited to 4 h, after which the experimenter gently woke the participant up and removed the eye-

tracking equipment. This was a precaution to avoid unwanted side effects such as drying of the right eye. Note that none of our participants reported negative effects, suggesting that longer measurement durations would be achievable. The videos generated by the pupillometer were processed with DeepLabCut[31,32] to detect pupil size, gaze, and some anatomical features of the eye. Since the eye was taped open without exceeding the participant's palpebral fissure during wake, there were some invalid video frames when the pupil was occluded by the eyelids. For 15 participants, pupil size could be calculated for at least 44.61% of the video recording (on average, mean ± SEM, 75.69 ± 5.66% valid samples, $n = 15$, see Supplementary Fig. 1B for distribution of valid pupil size measurement across sleep states). Two participant's pupils could not be accurately tracked for a large portion of the recording due to head movements resulting in shifting the pupillometer (69.82% invalid samples, $n = 1$) and tape displacement resulting in an insufficiently large palpebral fissure (70.37% invalid samples, $n = 1$).

### Pupil size and heart rate index dynamic changes in arousal levels at different moments of sleep

Arousal levels can be used to define sleep depth and have been shown to depend on the sleep macro-architecture[33,34]. To test this basic principle, we determined whether (i) pupil size (Fig. 1C), an established indicator of activity from arousal level-regulating systems in brainstem and basal forebrain regions during wakefulness[8,22,35] (ii) EEG spectral slope (Fig. 1D), a marker of cortical activity modulated by the aforementioned arousal level-regulating systems[33], and (iii) HR (Fig. 1D), a marker for cardiovascular activation, differed across sleep stages using linear mixed effect models and calculating post-hoc t-tests corrected for multiple comparisons using Benjamini-Hochberg correction. Pupil size, measured as the ratio between the pupil radius and the iris radius, was largest when being awake and significantly decreased for NREM stage 1 (N1), NREM stage 2 (N2), NREM stage 3 (N3), and REM sleep (Fig. 1F, $F(4, 57.43) = 168.90$, $p < 0.001$). The strongest changes were observed from wake to N1 and from N2 to N3, while pupil size was consistently small for N3 and REM sleep (see Supplementary Table 1). For a more detailed quantification of the distribution of pupil size across different sleep stages, see Supplementary Fig. 2A. Lower pupil size during NREM and REM sleep when compared to wake, in addition to the variable distribution of the size of the pupil within sleep stages is consistent with previous pupillometry findings in sleeping rodents, where NREM sleep is not subdivided into stages[19,36,37].

To assess whether pupil size can index arousal levels during sleep, we compared pupil size to spectral slope, an electrophysiological marker proposed to reflect the synaptic excitation-inhibition balance, with more negative spectral slopes indicating enhanced inhibition[38]. The spectral slope was calculated in 30-s segments in the 30−45 Hz range of the Fpz electrode[33]. Consistent with previous studies[33,39,40], we observed a significant decrease from wake to REM sleep in a near-linear fashion (Fig. 1G, $F(4, 68) = 41.37$, $p < 0.001$, Supplementary Table 2). Repeated measures correlation analyses revealed a significant positive correlation between pupil size and spectral slope across ($R(4264) = 0.34$, $p < 0.001$, 95% CID (0.31, 0.37)), see Supplementary Fig. 3 showing this within-individual association) and within sleep stages (Fig. 1I, all $R >= 0.18$, all $p < 0.001$, all 95% CID $\geq$ (0.11, 0.24)), except for REM sleep ($R(414) = 0.05$, $p = 0.291$, 95% CID (−0.04, 0.15)).

HR showed a U-shaped pattern from wake, through NREM sleep stages, to REM sleep (Fig. 1H, $F(4, 55.52) = 7.623$, $p < 0.001$), whereby HR was significantly lower in NREM sleep stages compared to wake (Fig. 1E, H and Supplementary Table 3). Relating HR and spectral slope, there was a weak yet significant correlation across sleep stages ($R(4264) = 0.09$, $p < 0.001$, 95%CID (0.056, 0.116)). Moreover, HR was significantly but weakly related to spectral slope during wake ($R(758) = 0.12$, $p = 0.001$, 95% CID (0.05, 0.19)) and REM ($R(414) = 0.14$,

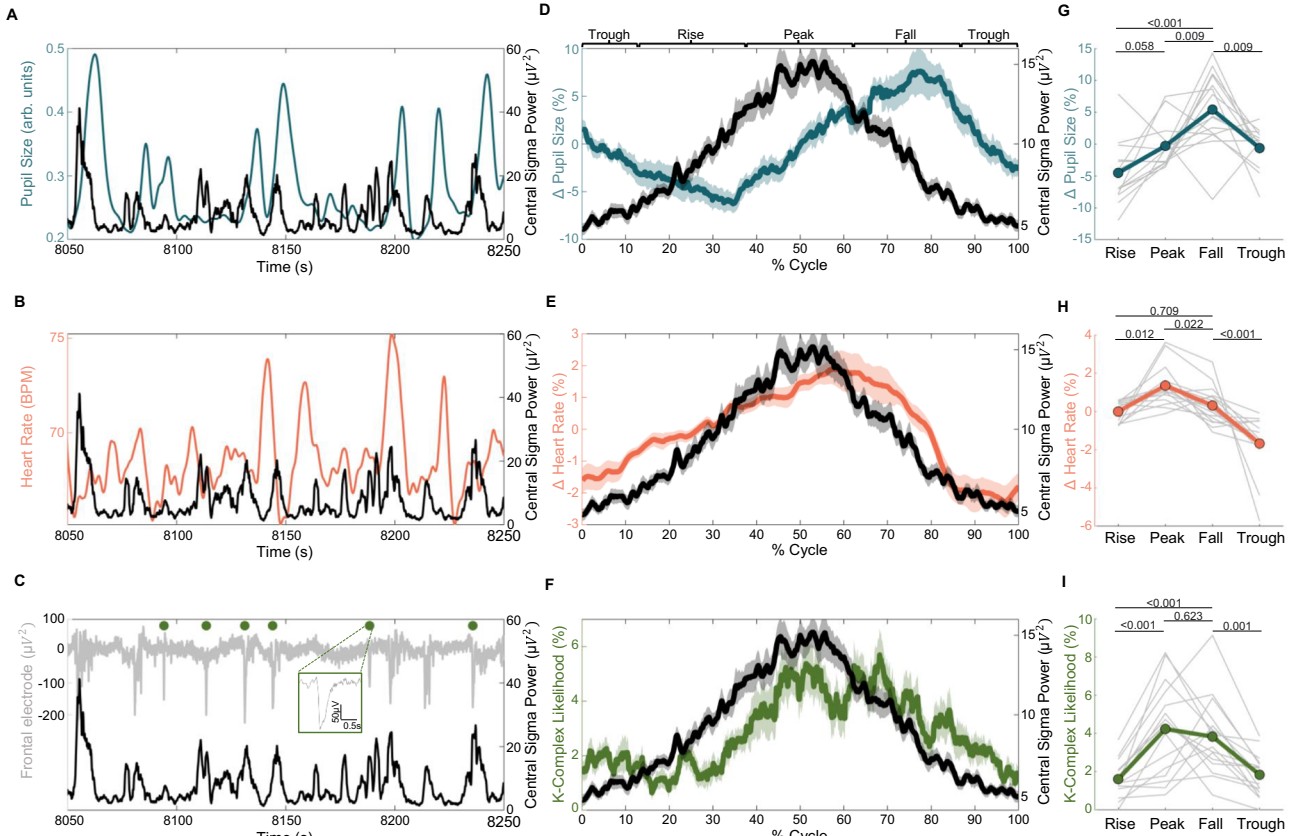

**Fig. 2 | Relationship between sigma power, pupil size, heart rate, and K-complexes. A–C** Individual example of sigma power fluctuations alongside the corresponding pupil size, HR in beats per minute (BPM), and K-complexes (green dots in C), respectively. The example window corresponds to the time in the gray boxes in the sleep recording from Fig. 1C and E. **D–F** Sigma power (black & right axis) with pupil size (blue & left axis in **D**), heart rate (red & left axis in **E**), and K-complex likelihood (green & left axis in F). All metrics are normalized to the timing of the troughs of detected spindle infraslow fluctuations (ISF). The time between troughs was renormalized in terms of cycle percentage. Centerline plots represent the mean across participants, and shadings around center lines represent the standard error of the mean (SEM). **G** Average pupil size ($F(3, 48.00) = 10.853$, $p < 0.001$), **H** heart rate ($F(3, 48.00) = 16.062$, $p < 0.001$), and **I** K-complex likelihood ($F(3, 48.00) = 17.602$, $p < 0.001$) for the rise, peak, fall, and trough of the normalized sigma power fluctuations. The data of each participant is depicted in gray. Statistical differences between the rise, peak, fall, and trough of sigma power ISFs are depicted, $p$-values are based on post hoc $t$-test and are adjusted for multiple comparisons using Benjamini-Hochberg correction. All statistical results are based on two-sided tests.

$p = 0.005$, 95% CID (0.04, 0.23)), but not significantly during NREM sleep stages (Fig. 1J, $R \leq 0.07$, $p \geq 0.215$).

In summary, our findings suggest that pupil size changes across sleep stages and appears to be linked to arousal levels reflected in spectral slope. HR also varied across sleep stages but seems to reflect the overall cardiovascular activation state, which is only weakly associated with arousal level markers.

**Distinct cortical and cardiovascular states are present across the arousal level spectrum of N2 sleep**

Spindle trains have been shown to exhibit infraslow fluctuations (ISFs) in rodents and humans[20,21,41]. Similar ISFs that are phase-locked to spindle clusters occur in the HR and the LC-mediated thalamic noradrenergic levels, where causal evidence shows that LC activity needs to be low for spindle trains to occur[16,18]. To test whether spindle ISFs also exist in humans, we calculated Cz sigma power (spindle frequency range of 12-16 Hz) during N2 (see Supplementary Fig. 4A for verifying the spindle ISF detection algorithm). We further tested whether similar infraslow dynamics are also observed for pupil size, HR, and K-complexes, NREM microevents representing highly synchronized bottom-up slow waves that potentially involve LC activation and the promotion of highly synchronized cortical activity[42,43] and has been argued to reflect arousal level fluctuations[14]. Figure 2A–C shows exemplary recordings of sigma power fluctuations (representing spindle ISF,

black lines) together with pupil size (Fig. 2A), HR (Fig. 2B), and K-complexes (Fig. 2C, green dots) during N2 sleep. All signals exhibited oscillations on infraslow timescales. Next, we applied a cycle-by-cycle analysis to sigma power (see "Methods" for details) and identified four phases marking sigma power's rise, peak, fall, and trough (Fig. 2D–F, black line). We then plotted pupil size (Fig. 2D), HR (Fig. 2E), and K-complex likelihood (Fig. 2F) as a function of the total spindle ISF cycle length (in %). All parameters exhibited significant modulations as confirmed by statistically comparing average pupil size, HR, and K-complex likelihood between the rise, peak, fall, and trough phase of the sigma power cycle using linear mixed effect models (all $F(3, 48) \geq 10.85$, all $p < 0.001$, Fig. 2G–I, Supplementary Tables 4–6).

Sigma power and pupil diameter appeared to fluctuate in a slightly phase-shifted manner (Fig. 2D). Cross-correlating pupil size and sigma power (Supplementary Fig. 4B) revealed that the two signals were shifted by approximately 30% of the cycle (corresponding to a 108-degree phase difference with $R = -0.26$, $p < 0.001$, see Supplementary Fig. 5A for significant lags). Pupil size was smallest during the rise of sigma power and reached its peak when sigma power was falling (Fig. 2G). Note that pupil size was largest during the fall of sigma power across all phases ($p < =0.009$).

Contrary to pupil size, HR and K-complex likelihood fluctuations tended to occur phase-locked to sigma power fluctuations (Fig. 2E, F). A cross-correlation between sigma power and HR revealed a delay of

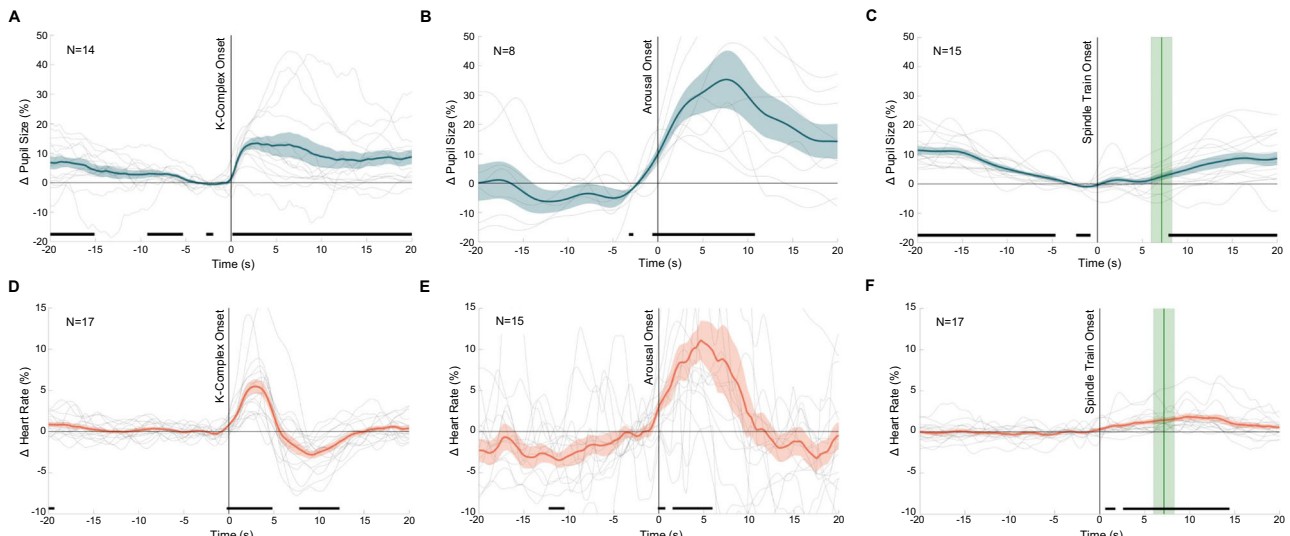

**Fig. 3 | Pupil size and instantaneous heart rate dynamics of N2 events. A–C** Pupil size color-coded in blue and normalized to the 5 s prior the detected event: K-complexes, sleep arousals, and spindle trains (respectively).**D–F** Heart rate color-coded in orange and normalized to the 5 s prior to the detected event: K-Complexes, spindle trains, and sleep arousals (respectively). **C**, **F** Green vertical shading in top panels reflects the mean ± 1 SEM of the median spindle train end across participants. The center lines represent the group mean, and shading around the center lines is the SEM. Gray lines are the mean response of each participant. Black horizontal lines mark significant differences from the 5 s baseline prior to each event ($p < 0.05$). $p$-values are based on a post hoc $t$-test and are adjusted for multiple comparisons using Benjamini-Hochberg correction. All statistical results are based on two-sided tests.

the HR fluctuation of approximately 5% (corresponding to an 18-degree phase difference, $R = 0.26$, $p < 0.001$, see Supplementary Fig. 5B for significant lags) and HR was significantly higher during the peak phase of sigma power than during the rise or fall ($p <= 0.022$). K-complexes were significantly less likely to occur during the rise of sigma power than during the peak and fall of sigma power (both $p < 0.001$).

In summary, we show for the first time in humans that fluctuations in pupil size, HR, and K-complex likelihood are linked to spindle ISFs. Spindle ISFs are an EEG marker that has been previously shown to be linked to LC-mediated arousal levels during NREM sleep in rodents, i.e. spindle occurrence is low when LC-NA levels are high and vice versa[16,18].

**Pupil size and heart rate fluctuations reveal distinctly different dynamics across N2 events**

Next, we investigated how pupil size and heart rate relate to three specific events in N2 sleep micro-architecture that hallmark changes in arousal levels, i.e., K-complexes, spindle clusters, and sleep arousals as defined in the American Academy of Sleep Medicine (AASM) criteria (see Supplementary Fig. 6D for an overview of pupil size and heart rate dynamics)[17,20,44–46]. According to these criteria, sleep arousals are characterized by abrupt changes in EEG frequency, suggestive of an awake state, which in REM sleep must be accompanied by increases in EMG amplitude[17,47]. In essence, sleep arousals capture snapshots of brief arousal level changes that are elevated enough to manifest as EEG-derived cortical activity. Although sigma power provides an indirect but continuous readout of spindle activity, discretizing the Cz signal into spindle clusters provides a uniform timescale in addition to a more sensitive and precise readout of strong spindle activity. It is worth noting that not all participants exhibited each event type together with valid pupil size or HR measurements during epochs with unperturbed sleep (Fig. 3). To evaluate if pupil size and HR are different in relation to these sleep-microarchitecture events and therefore distinct from the overall N2 state, we investigated pupil and HR dynamics at each N2 event type and used t-tests corrected for multiple comparisons using Bejamin-Hochberg correction. We also calculated the derivative of pupil size during these events as it has been found to

be tightly linked to noradrenergic modulatory systems and to cortical activity (Supplementary Fig. 7)[8,48].

For K-complexes (Fig. 3A), we observed a clear pupil dilation within the 10 s after the K-complex onset with an average increase of $13.49 ± 2.21\%$ from baseline. Interestingly, systematic changes in pupil diameter are revealed that appeared to initiate at the estimated onset of the K-complex but did not return back to baseline within the 20 s after the onset. This pupil size increase at onset was further highlighted by a significant positive change in the derivative of pupil size (Supplementary Fig. 7A). Figure 3A also shows that average pupil size gradually decreased to a minimal level before the dilation started (this is supported by the significant negative derivative of pupil size at time $t = −5$ in Supplementary Fig. 7A). Furthermore, HR revealed a biphasic response to K-complexes, reflected in an initial increase of up to $5.49 ± 0.87\%$ from baseline ($p < 0.05$), followed by a decrease of up to $2.84 ± 0.48\%$ below baseline ($p < 0.05$, Fig. 3D). This biphasic HR response to K-complexes aligns with previous findings[46].

Sleep arousals (Fig. 3B), as defined by AASM guidelines are brief perturbations to sleep which are characterized by large and sudden increases in the theta, alpha, or beta frequencies in the EEG power spectra, which can be accompanied by increased EMG activation. We found sleep arousals to be accompanied by a pupil dilation that typically started slightly before the determined sleep arousal onset and reached a significant increase of up to $35.37 ± 9.88\%$ compared to baseline, around 7 s after sleep arousal onset ($p < 0.05$). This pupil dilation was descriptively larger than the dilations observed during K-complexes but also more variable across participants as the derivative of pupil size did not significantly deviate from zero (Supplementary Fig. 7B). When comparing pupil size dynamics between K-complexes and sleep arousals, statistics did not survive correction for multiple comparisons. Similarly, HR increased by $11.18 ± 2.31\%$ in parallel with sleep arousals ($p < 0.05$, Fig. 3E) which is descriptively but not statistically larger in magnitude than HR increases observed during K-complexes (Fig. 3D, E).

Spindles (Fig. 3C) and their clustering into "trains" occur during low LC-NA activity (low arousal level) and have been argued to play a sleep-protecting role[44,49]. In line with the pupil size fluctuations during spindle ISFs (Fig. 2D, G), pupil size exhibited a U-shaped pattern during

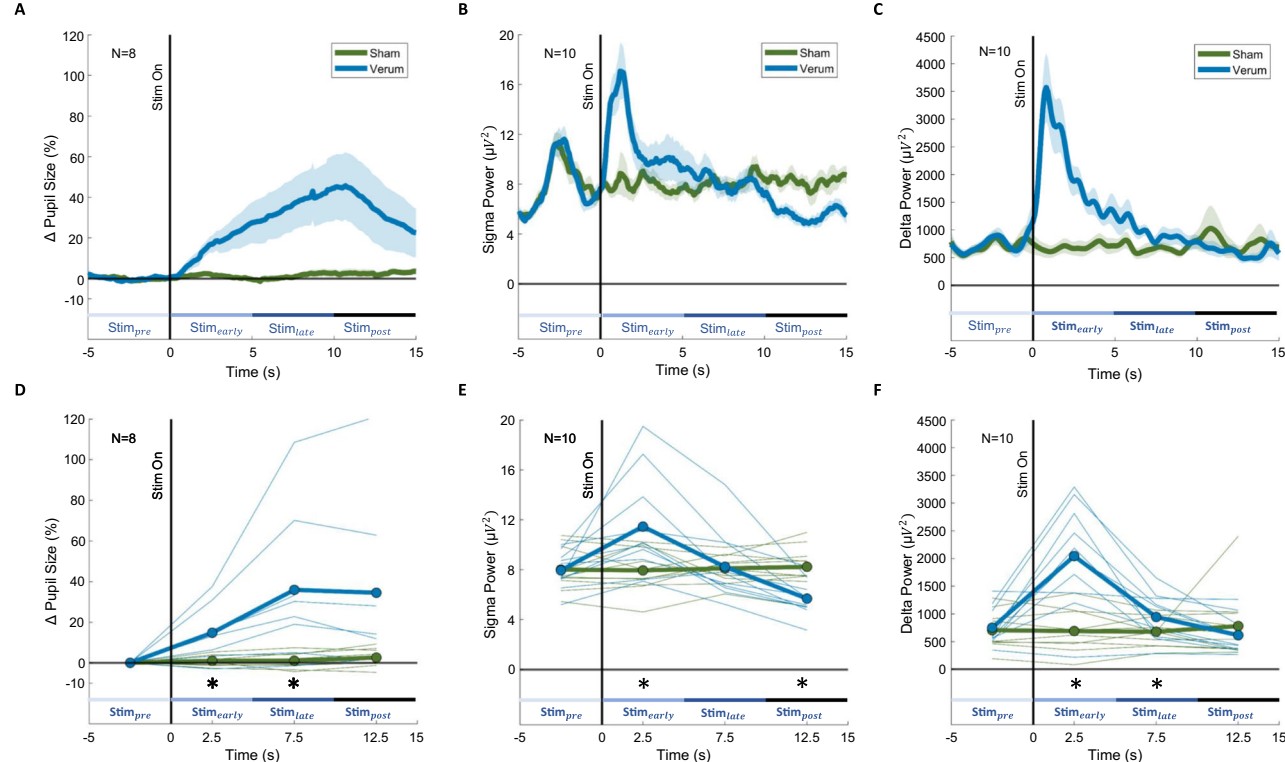

**Fig. 4 | Pupil and EEG responses to auditory stimulation.** We applied a 10-s stimulation condition of 1 Hz rhythmic auditory stimulation at 45 dB (verum) at varying arousal levels and a control condition where no stimuli were applied (sham). **A**–**C** Pupil size, sigma power, and delta power relative to the 5 s prior to the detection of verum (blue) and sham (green) stimulation windows. Center lines represent the mean, and shading around center lines represent SEM. Time zero is the start of the stimulation window. The stimulation window start times were marked by the stimulation-algorithm independent of whether sham or verum followed. They only differ in terms of whether real tones were played or not. **D**–**F** Pupil size, sigma power, and delta power of **A**–**C**, respectively, during verum (blue) and sham (green) averaged every 5 s from time zero onwards for each participant. Black stars highlight significant differences ($p < 0.05$) between verum and sham for a given 5 s region. *p*-values are based on a post hoc *t*-test and are adjusted for multiple comparisons using Benjamini-Hochberg correction. **D** Pupil in verum condition was significantly larger than sham during $Stim_{early}$ ($13.77 \pm 4.81\%$, $p = 0.046$) and $Stim_{late}$ ($34.87 \pm 12.90\%$, $p = 0.046$). **E** Sigma power in verum condition was significantly larger during $Stim_{early}$ relative to sham ($3.52 \pm 1.30\ \mu V^2$, $p = 0.036$) and significantly lower during $Stim_{post}$ relative to sham ($-2.55 \pm 0.46\ \mu V^2$, $p = 0.001$). **F** Delta power in verum condition increased during $Stim_{early}$ ($1'353 \pm 271.54\ \mu V^2$, $p = 0.002$) and $Stim_{late}$ ($265.95 \pm 78.29\ \mu V^2$, $p = 0.001$) when compared to sham and then during $Stim_{post}$ returned then back to $Stim_{pre}$ values ($-165.23 \pm 135.99\ \mu V^2$, $p = 0.255$). See Supplementary Table 8 for a list of participants of the participants used in this analysis. All statistical results are based on two-sided tests.

the 40-s period surrounding spindle trains. Prior to the onset of the spindle train, there was a significant $11.51 \pm 1.46\%$ decrease in pupil size compared to baseline ($p < 0.05$) which corresponded to a significant negative derivative of pupil size during this time period (Supplementary Fig. 7C). Pupil size remained relatively small throughout the spindle train and only after did it return to values observed 20 s prior to the onset of the spindle train ($p < 0.05$), as reflected by the significant positive derivative of pupil size after the spindle trains (Supplementary Fig. 7C). We found that HR started to slowly increase at the onset of spindle trains and remained $1.851 \pm 0.42\%$ higher than baseline beyond the end of most spindle trains ($p < =0.05$, Fig. 3F).

To shed light on the pupil size and HR dynamics surrounding K-complexes, sleep arousals, and spindles, we compared the baseline of each event to the average pupil size ($F(3, 39.30) = 3.68$, $p = 0.020$) and HR ($F(3, 40.03) = 6.34$, $p = 0.001$) during periods of N2 where tones were not played (general N2 levels). Pupil size was not significantly different to general N2 levels prior to K-complexes ($t(41.8) = 2.22$, $-8.45 \pm 2.33\%$, $p = 0.064$, 95% CID ($-0.05$, $0.003$)), spindle trains ($t(41.5) = -2.30$, $-8.34 \pm 2.75\%$, $p = 0.064$, 95% CID (($-0.05$, $0.002$)), and prior to sleep arousals ($t(43.3) = 0.38$, $0.95 \pm 4.94\%$, $p = 0.7061$, 95% CID ($-0.03$, $0.04$)). Conversely, HR was significantly higher prior to sleep arousals ($t(43.4) = 3.20$, $3.50 \pm 1.59\%$, $p = 0.008$, 95% CID ($0.58$, $4.62$)) but was not significantly different from general N2 levels prior to K-complexes ($t(43.2) = -1.25$, $-1.18 \pm 0.46\%$, $p = 0.381$, 95% CID ($-2.04$,

$0.67$)) and spindle trains ($t(43.2) = -0.89$, $-0.89 \pm 0.71\%$, $p = 0.381$, 95% CID ($-1.84$, $0.88$), Supplementary Table 7).

In summary, before K-complexes and spindle trains the pupil constricted while HR remained stable. After all three events, the pupil dilated while HR showed less pronounced responses (Supplementary Fig. 6).

## Cortical response to sensory stimulation varies with arousal levels

Finally, we investigated whether arousal levels, as indexed by pupil size, affect cortical response to external stimuli. We applied auditory stimulation (verum) at varying arousal levels, and a control condition where tones were not played (sham, Fig. 4). The onset of verum and sham was conditional on the detection of the ascending phase of a slow wave during NREM sleep or on the detection of an increase or a decrease in sigma power (see Methods for details and Supplementary Fig. 8 for polysomnography data of exemplary trials). To statistically compare verum versus sham conditions, we normalized the data to $Stim_{pre}$ and averaged within three-time bins (i) $Stim_{early}$ (first 5 s of the auditory stimulation), (ii) $Stim_{late}$ (last 5 s of the auditory stimulation), (iii) $Stim_{post}$ (5 s bin after auditory stimulation was switched off) and applied paired t-tests between verum and sham for each time bin and corrected for multiple comparisons using Benjamini-Hochberg correction (Fig. 4; Methods). We found that the pupil showed a sharp

**A**

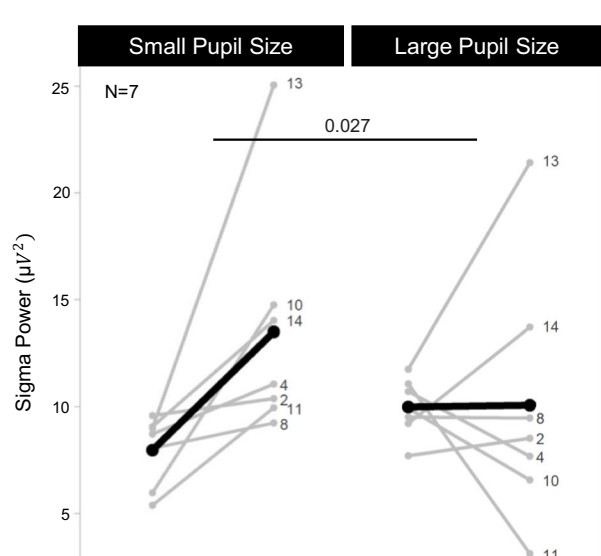

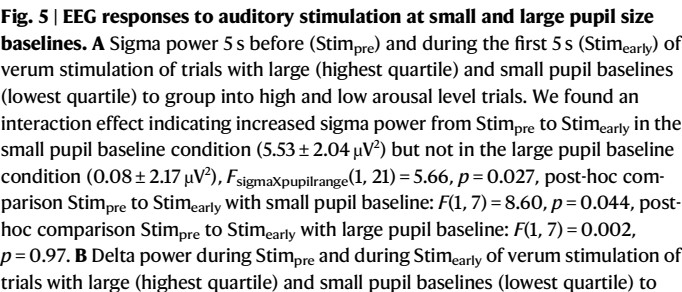

**B**

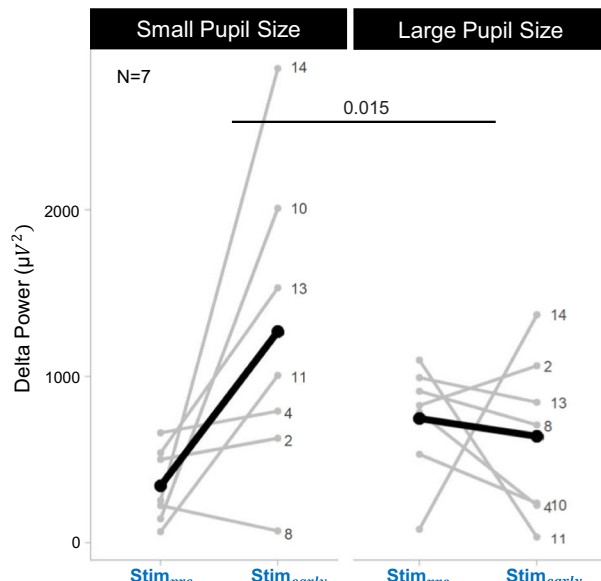

**Fig. 5 | EEG responses to auditory stimulation at small and large pupil size baselines. A** Sigma power 5 s before (Stim$_{pre}$) and during the first 5 s (Stim$_{early}$) of verum stimulation of trials with large (highest quartile) and small pupil baselines (lowest quartile) to group into high and low arousal level trials. We found an interaction effect indicating increased sigma power from Stim$_{pre}$ to Stim$_{early}$ in the small pupil baseline condition ($5.53 \pm 2.04 \, \mu V^2$) but not in the large pupil baseline condition ($0.08 \pm 2.17 \, \mu V^2$), $F_{sigmaXpupilrange}(1, 21) = 5.66$, $p = 0.027$, post-hoc comparison Stim$_{pre}$ to Stim$_{early}$ with small pupil baseline: $F(1, 7) = 8.60$, $p = 0.044$, post-hoc comparison Stim$_{pre}$ to Stim$_{early}$ with large pupil baseline: $F(1, 7) = 0.002$, $p = 0.97$. **B** Delta power during Stim$_{pre}$ and during Stim$_{early}$ of verum stimulation of trials with large (highest quartile) and small pupil baselines (lowest quartile) to

group into high and low arousal level trials. We found an interaction effect indicating increased delta power from Stim$_{pre}$ to Stim$_{early}$ in the small pupil baseline condition but not in the large pupil baseline condition $F_{deltaXpupilrange}(1, 28) = 6.77$, $p = 0.015$, post-hoc comparison Stim$_{pre}$ to Stim$_{early}$ with small pupil baseline: $F(1, 14) = 7.58$, p = 0.031, post-hoc comparison Stim$_{pre}$ to Stim$_{early}$ with large pupil baseline: $F(1,14) = 0.25$, p = 0.623. The numbers next to the gray lines correspond to the participant number. The stimulation protocol V0 was used in 2 and 4, and V1 for the rest. The horizontal black lines connecting pupil size ranges are the interaction effects between pupil size range and Stim$_{pre}$-Stim$_{early}$ power values. *p*-values are based on a post hoc *t*-test and are adjusted for multiple comparisons using Benjamini-Hochberg correction. All statistical results are based on two-sided tests.

dilation during Stim$_{early}$ and Stim$_{late}$ that was only present during verum stimulation (Fig. 4A, D). This dilation reached its peak shortly after the end of the 10 s stimulation window, upon which pupil size gradually decreased again. Sigma power, being associated with an increased likelihood of sleep spindle occurrence, significantly increased during verum Stim$_{early}$ when compared to sham, which is typically observed when slow wave amplitude is increased by tones[50,51]. However, after the initial peak, sigma power decreased quickly and was significantly suppressed during Stim$_{post}$ relative to sham (Fig. 4B, E). In parallel, delta power representing slow wave activity increased during Stim$_{early}$ and Stim$_{late}$ when compared to sham, and then during Stim$_{post}$ returned to Stim$_{pre}$ values (Fig. 4C, F).

We further investigated whether pupil size, as an indirect marker of arousal levels prior to playing a tone, modulates the response of sigma and delta power to auditory stimulation. Therefore, we separated N2 verum trials where pupil size during Stim$_{pre}$ was large (highest quartile, representing "high" arousal level) versus small (lowest quartile, representing "low" arousal level) and compared the sigma and delta power 5 s prior to stimulation onset (Stim$_{pre}$) with the first 5 s of stimulation (Stim$_{early}$) (Fig. 5A, B; Methods). Linear mixed effect models revealed a significant difference between large and small pupil size during Stim$_{pre}$ to general N2 levels ($F(2, 14.00) = 47.06$, $p < 0.001$; see Supplementary Fig. 9). Post-hoc *t*-tests corrected for multiple comparisons revealed that large pupil size during Stim$_{pre}$ was greater than general N2 levels ($t(16.3) = 3.37$, $18.54 \pm 5.50\%$, $p = 0.004$, 95% CID (0.01, 0.08)) and small pupil size baselines were smaller than general N2 levels ($t(16.3) = -5.527$, $-25.21 \pm 3.40\%$, $p < 0.001$, 95% CID ($-0.11$, $-0.04$), Supplementary Fig. 9). We then investigated the associations between baseline and evoked pupil dilation and found main effects of pupil size at baseline ($F(1, 21) = 54.94$, $p < 0.001$) and evoked pupil

dilation ($F(1, 21) = 9.60$, $p = 0.005$). We found no interaction effect between baseline and evoked pupil dilation, suggesting that the evoked pupil dilation was not significantly different at different baselines ($F_{baselineXdilation}(1, 21) = 0.042$, $p = 0.829$; see Supplementary Fig. 10). There was an increase in sigma power from Stim$_{pre}$ to Stim$_{early}$ in the small pupil baseline condition but not in the large pupil baseline condition (Fig. 5A). Furthermore, there was an increase in delta power from Stim$_{pre}$ to Stim$_{early}$ in the small pupil baseline conditions ($928.67 \pm 381.79 \, \mu V^2$) but not in the large pupil baseline conditions ($-105.90 \pm 277.54 \, \mu V^2$, Fig. 5B). When we included all quartiles into which the N2 verum trials were separated, the aforementioned relationship of increasing evoked response in power with decreasing pupil size at baseline became nonlinear (Supplementary Fig. 11). Increased evoked response in sigma and delta power occurred in intermediary quartiles but not as robustly as in the small pupil baseline conditions.

In summary, our findings show that auditory stimulation during sleep induces a pupil response upon stimulation. Furthermore, we demonstrate distinct cortical responses to auditory stimulation at varying arousal baseline levels as indexed by pupil size prior to stimulation onset.

## Discussion

In the present study, we monitored pupil size during human sleep using a low-dose portable infrared system and compared it to cortical and cardiovascular events measured via polysomnography and ECG. We demonstrate that pupil size dynamics relate to macro- and microarchitecture of sleep and predict the brain's response to auditory stimuli. Pupil size dynamics during human sleep not only correlate with cortical markers of arousal levels but are also in line with arousal level dynamics reported in rodent sleep. Therefore, pupil size

dynamics might represent a non-invasive way to track arousal levels in human sleep.

We showed in humans that pupil size decreases significantly from wakefulness to deep NREM sleep, reaching an equally small diameter in N3 and REM sleep. Similar pupil size changes across sleep states have previously been reported in rodent models[19,36,37]. Here, we reproduced the pronounced decrease from wakefulness to NREM sleep (N1 and N2) and then to N3 and REM sleep. Note that there is no currently established subdivision of rodent NREM sleep into stages. The small pupil size that characterizes both REM sleep and the N3 stage of NREM sleep has remained undetected until now. For a more direct comparison with pupil size changes across sleep states in rodent models, we pooled NREM sleep stages together (Supplementary Fig. 2B). This revealed a coarser distribution in pupil size during NREM which largely overlaps with pupil size values during REM and minutely with wakefulness. It is remarkable that similar to pupil diameter distributions, LC activity ranges in rodent NREM sleep also overlap with the ones of REM sleep and of wakefulness[52]. The positive skewness of the pooled data, as a result of N3 samples, is also visible in rodent sleep[19,36,37]. It is, therefore, tempting to speculate that the low pupil size values in NREM rodent sleep is a distinct brain state with analogous properties to human N3 sleep. Furthermore, the distribution of the data suggests that pupil size variations may reflect the dynamic range of arousal level fluctuations within the noradrenergic system described for rodent sleep.

In the present study, observed changes in pupil size were paralleled by a decrease in spectral slope, an electrophysiological marker of arousal level[33], as NREM sleep deepened. Spectral slope levels decreased nearly linearly across NREM sleep stages. These findings suggest that pupil dynamics capture pronounced changes in the vigilance state. Unlike pupil dynamics, spectral slope did further decrease from N3 to REM sleep, suggesting that spectral slope and pupil diameter might reflect slightly different physiological mechanisms during REM sleep even though it cannot be ruled out that further pupil constriction was prevented due to a floor effect. HR, on the other hand, showed little to no association with the spectral slope, across sleep stages and also within NREM and wake. This suggests that HR variations may be too moderate to capture arousal level fluctuations or that this measure seems to be less closely linked to arousal level dynamics indexed by the spectral slope. However, other ECG-derived measures not investigated here may further elaborate the role of autonomic nervous system activity in pupil size dynamics during sleep[53].

We observed infraslow fluctuations (ISFs) of sigma power (12-16 Hz) in N2, which reflects sleep spindle probability (Supplementary Fig. 4A). Our additional finding is that human pupil size also exhibits ISFs so that pupil diameter is small before sigma power increases (Fig. 2A, D, G). This is in line with previous work in mice showing a similar relationship between low pupil size and high sigma power[19,20]. Furthermore, it has been shown that also LC-NA activity exhibits pronounced ISFs during NREM sleep and that high LC-NA activity is associated with a decreased occurrence of sleep spindles. This may be due to the depolarization of thalamocortical and thalamic reticular neurons, which delays their re-engagement in sleep-spindle generation[16]. Interestingly, experimentally manipulating LC-NA activity has a causal influence on sleep spindle occurrence: inhibiting LC-NA activity facilitates spindles while activating LC-NA suppresses them[16,18]. It has also been tested whether the response to external stimuli would systematically vary between different phases of these ISFs, and research in mice has revealed that presenting stimuli during high LC-NA activity/low sleep spindle occurrence caused an increased likelihood of sleep arousals and awakenings while these were substantially less likely when stimuli are presented during low LC-NA/high sleep spindle occurence[20,25,44].

During the infraslow fluctuations of sigma power in N2, we found HR, a marker commonly used to infer autonomic activity during sleep,

to be high while sigma power was high (Fig. 2B, E, H, Supplementary Fig. 4C). This is in line with research in humans but, interestingly, not with findings in mice for which heart rate was low during high sigma power[19,20]. Lecci et al. suggested that the delay in the relationship between HR and sigma power may be related to anatomical and physiological differences between the neural coupling to the heart in both species. In general, phase relationships to sigma power variations need to be interpreted cautiously because they represent a low-pass-filtered version of the neural activity of the LC[52], most likely because of the slow NA signaling mechanisms in thalamic neurons[16]. Our findings and previous literature suggest a tight coupling between pupil-indexed size, our indirect marker of arousal levels, and heart rate to sigma power at an infraslow timescale (Supplementary Fig. 4B, C). However, there are no studies to our knowledge that have investigated pupil size or HR fluctuations during individual spindle trains, a more sensitive and temporally precise approach to capture spindle activity underlying sigma power (Supplementary Fig. 4A)[54]. Using this approach, we found dynamic relationships between pupil size and heart rate to spindle clusters that went unnoticed in the more conventional correlational analysis (Supplementary Fig. 6C, D). Pupil size also constricted before the occurrence of spindle clusters and dilated after, as indicated by the significant pupil size derivative (Supplementary Fig. 7C). This was perhaps as a result of the reduced arousal levels mediated through reduced noradrenergic activity suggested to be required for spindles to occur[44]. Conversely, HR remained largely unaffected prior to the occurrence of sleep spindles.

We further established how pupil dynamics evolve around other microstructural events typically observed during sleep, such as K-complexes and sleep arousals. Similarly to spindle clusters, pupil size constricted before the onset of a K-complex, but this was not the case for sleep arousals (Fig. 3A-C). Therefore, a reduction in arousal levels seems to be tightly linked to the likelihood for sleep-related cortical events such as K-complexes and sleep spindles to occur, a relationship that can be captured by measuring pupil size and that is also evident in the dynamics of LC activity prior to an activity surge in rodents[52]. Similarly to spindle clusters, HR remained largely unaffected prior to the occurrence of K-complexes but showed an increase following the onset of these events. Therefore, HR seems to be less closely linked to arousal level dynamics prior to sleep spindles and K-complexes per se but may rather be used to classify the autonomic response to such events[45]. In rodents, Eschenko et al. reported increased LC firing at the Down-to-Up state transition of slow waves, potentially leading to the synchronized depolarization of a large number of cells, which then influences the cortical networks generating this state transition[55]. In addition, Osorio-Forero et al. show that relatively small LC activity surges lead to a phasic increase in low-frequency power, whereas larger ones cause a short bout of wakefulness from NREM. This is evidence in favor of a functional analog of human K-complexes in rodents that reflects a graded arousal level in LC activity[52]. Here, we specifically focused on K-complexes and found rapid pupil dilations at their onset, pupil size dynamics that have been associated to phasic activity of noradrenergic axons[8]. Along these lines, Osorio-Forero et al. reported that transient LC activity increased delta power, possibly reflecting K-complexes that remain undefined in sleeping rodents[52]. Our results indicate that K-complexes involve a significant pupil dilation possibly reflecting transient LC activity time locked to these events. Interestingly, sleep arousals were also accompanied by pupil dilation and have previously been reported to be accompanied by bursts in LC activity[18]. However, this dilation was not as robust as that reported at the onset of K-complexes. This discrepancy may be due the unclear temporal alignment between pupil size dynamics and the detection of sleep arousal onsets as it is based on thresholding of changes in EEG power bands whereas with K-complexes it is based more precisely on the event-related potential of the EEG signal.

Assuming that different pupil levels track different arousal levels, we would expect that the cortical response to auditory stimulation differs depending on when these stimuli are played as a function of pupil size. According to previous research, we found auditory stimulation to increase delta and sigma power dynamics[50,56–62]. These delta responses are believed to involve enhanced slow waves or K-complexes, while sigma responses involve enhanced spindle activity[63,64]. However, this increase in delta and sigma power was indeed not present when pupil size was high (largest quartile) prior to stimulation onset (Fig. 5 and Supplementary Fig. 11). Therefore, a minimum pupil size might be a prerequisite for the induction of sensory-evoked delta and sigma enhancement, with the optimal evoked response potentially occurring at low arousal levels. This is in contrast to inverted U-shaped relationships between pupil-indexed arousal levels and neuromodulation that have been observed during wakefulness in rodents[65,66], and therefore adds additional complexity to our current understanding of neuromodulation across the entire arousal level continuum[67]. In rodents, Hayat et al.[25] showed that tonic LC activity levels were higher before tones that led to sensory-evoked NREM sleep perturbations compared to tones resulting in maintenance of NREM sleep[25]. Importantly, in both cases, the tones induced a phasic firing of LC time-locked to the tone onset, but only high prior arousal levels resulted in sleep awakenings characterized as wake-like EEG flattening (without dominant theta) lasting for at least 3 s. In this regard, our results might point to the idea that during lower arousal levels, as indicated by everything below high pupil size here, slow oscillations might be induced as a consequence of the short phasic firing of LC which may act as sleep-protective. K-complexes have been suggested by Colrain et al. to be sleep protective inhibiting arousal level surges and to promote cortical processing, but this is still debated[64,68]. However, our stimulation protocol did only involved low volume sounds that are generally not inducing sleep perturbations or changes in sleep architecture[56]. Overall, our results indicate that the arousal level state prior to stimulation onset significantly influences the effectiveness of auditory stimulation to enhance K-complexes or slow waves in the form of delta power, as well as spindle activity in the form of sigma power. Consequently, pupil size as a proxy for arousal level dynamics may essentially improve current auditory stimulation protocols to enhance slow waves and spindles more consistently. Specifically, we could monitor pupil size to optimize existing sleep stimulation protocols to enhance slow waves and spindles more effectively or to develop protocols for individuals that have so far showed weak or no responses to auditory stimulation[69]. Tracking the likelihood of evoking slow waves or spindles upon different sound volumes across pupil level fluctuations may help improve auditory stimulation protocols used in sleep. While finer binning of pupil size before stimulation windows could have provided further insights into the relationship between sensory-evoked responses and specific arousal levels, the auditory stimulation analysis was constrained by the number of trials and participants with valid pupil data during stimulation windows. Monitoring pupil size online and applying sensory stimulation based on ongoing pupil size dynamics in a larger sample could provide a more representative sample of evoked response across all arousal levels of NREM. Future studies could use such an approach to identify optimal arousal levels to maximize target responses while mitigating the likelihood of awakenings. Our approach holds the potential to unlock a research branch within the field of sleep, as it may enable us to track arousal levels during human sleep. This breakthrough allows for fresh fundamental discoveries regarding the interaction between arousal levels and sleep function, potentially offering indirect insight into the LC-NA system during human sleep. Although the current literature in other animal species predominantly supports pupil size to be an indirect readout of LC-mediated arousal levels during sleep[25], other mechanisms may be modulating sleep-related pupil size fluctuations directly or indirectly, such as parasympathetically dominant autonomic cholinergic activity, or hypothalamic orexin neurons signaling through the LC[8,10,19]. Moreover, our findings could have potential clinical implications in the diagnosis of sleep disturbances. Perturbed arousal levels and LC-function have been closely associated with insomnia, stress-related disorders, and neurodegenerative diseases like Alzheimer's and Parkinson's, which often manifest with disrupted sleep patterns[70–73]. By employing our approach, clinicians could directly diagnose whether abnormally elevated arousal levels underlie perturbed sleep in these patients, thus motivating alternative treatment approaches. Recent advancements in estimating pupil size behind closed eyelids hold promise for the field of sleep research, which could build on our findings provided they can accurately capture subtle pupil size dynamics during sleep[74].

In conclusion, this study highlights the potential of pupil size as a non-invasive marker of arousal levels during human sleep. Pupil size dynamics correspond to sleep macrostructure, exhibit ISFs during NREM sleep, and change transiently after naturally occurring microevents like K-complexes and sleep arousals. Interestingly, a small reduction in pupil size seems to be a prerequisite for K-complexes and spindles to occur. Similarly, a small pupil size prior to auditory stimulation influences its effectiveness in enhancing delta and sigma power. These findings provide insights into the interplay between arousal levels and sleep, opening research avenues to potentially further understanding disorders associated with abnormal arousal levels. Pupil size monitoring, therefore, holds promise as an innovative tool in both human sleep research and clinical practice.

## Methods

### Participants

In total, 18 participants were recruited for this study. One participant was excluded because of predefined exclusion criteria, as they reported an elevated level of eye dryness (score greater than 12 in the Ocular Surface Disease Index questionnaire[75], 0 to 12 representing normal levels of eye dryness). The 17 included participants (10 female, mean ± sd age; 29.83 ± 8.69 years) were non-smokers, reported a regular sleep-wake rhythm, and had a body-mass index between 17–30. Participants reported no presence of psychiatric/neurological diseases, sleep disorders, or clinically significant concomitant diseases. The study was approved by the cantonal Ethics Committee Zurich (reference number: KEKZH, BASEC2022-00340) and conducted in accordance with the declaration of Helsinki. All participants provided written informed consent prior to study participation and received monetary compensation (CHF 70).

### Experiment procedure

A graphical overview of the experimental procedure is illustrated in Fig. 1A. Prior to enrollment, a telephone screening was conducted to explain the procedure and address questions. On the day of testing, participants answered questionnaires about demographics, health status, handedness, sensitivity to noise, eye health, sleep habits, sleep quality, and daytime sleepiness (not reported here). Thereafter, gold electrodes (Genuine Grass electrodes, Natus Medical Inc., Pleasanton, US) were placed to record EEG (Fpz, Cz, O1, O2, M1 (Reference), according to 10-20 system), chin EMG, EOG, and ECG. All channel impedances were kept below 20kΩ. Etymotic insert earphones (Etymotic Research Inc., ER 3 C) were then placed in the ear canals and taped to prevent dislodgment during the night. Subsequently, participants were laid in a supine position on the bed where the sides of the pillow were wrapped with folded towels to restrict the head from side movements. The right eye remained open by affixing four pieces of Hypafix tape (BSN Medical GmbH & Co KG, Hamburg, Germany) to the upper eyelid: three strips were placed on the forehead and attached to a fourth strip that was folded outward and positioned onto the upper eyelid. A single, broad strip of tape was used on the lower eyelid (Fig. 1B). To prevent eye dryness, vitamin A eye ointment was applied

on the eye immediately before the eye was covered by a transparent eye bandage (PRO Optha S, Lohmann & Rauscher International GmbH & Co. KG, Rengsdorf, Germany). Prior to the application, baby soap was used to clean the plastic dome inside the eye bandage, to minimize the accumulation of water condensation. Finally, the eye tracker goggles (Pupil Core, Pupil Labs GmbH, Berlin, Germany) were placed as shown in Fig. 1B. Lights out (approximately 10–11 pm) occurred as close as possible to the time participants reported going to bed.

With the exception of the small infrared light source on the goggles, all light sources from the apparatus were covered using black tape or blackout curtains. Once the experimenter turned off the lights and left the sleep lab, participants made 3 up-and-down and 3 left-and-right eye movements that indicated the start of the participant's sleep window onset of 7.5 h. During the complete sleep window, we recorded polysomnography and ECG using a BrainAmp ExG amplifier (BrainProducts GmbH, Gilching, Germany) through OpenVIBE[76] at a sampling rate of 500 Hz. Additionally, the eye tracker video stream was recorded using the manufacturer's software, Pupil Capture. The experimenter on call entered the sleep lab up to 4 h after taping the eye open to remove the goggles, eye bandage, all tape stripes, and the head restricting towels. In the morning, the experimenter quietly entered the sleep lab, stopped all data acquisition devices, and woke the participant up. The session ended with a second round of sleep quality and mood questionnaires and reimbursement (see Supplementary Fig. 12 for a summary of questionnaires).

### EEG analysis and sleep scoring

All data prior to the moment of the horizontal and vertical eye movements (visually detected in the EOG electrodes) marking the beginning of the sleep window were excluded from further analysis. Then, EEG data was down-sampled to 200 Hz using the EEGLAB[77] toolbox function *pop_resample* in MATLAB (R2019a, MathWorks Inc., Natick, MA). Thereafter, the automatic sleep staging tool YASA[78] was used to predict sleep stages in 30 s windows. Sleep stage predictions with less than 0.6 confidence were visually inspected using the sleep stage visualizer toolbox Visbrain[79] and manually corrected by an expert scorer following the standard AASM scoring guidelines[17]. Artifacts were rejected based on a semiautomatic artifact removal procedure that was described previously[80]. This method involved automatically excluding 30-s epochs of data if their power values deviated from the background power. The background power was calculated using a sliding mean in two specific frequency ranges: 0.75–4.5 Hz and 20–30 Hz. Epochs with power values clearly different from this background measure were excluded. To determine instantaneous delta and sigma power, respectively, Fpz data was bandpass filtered in the 0.5 to 2 Hz (delta) frequency range, and Cz data in the 12 to 16 Hz (sigma) frequency range using MATLAB 2nd order Butterworth filters. Then, the Hilbert transform of each filtered signal was used to obtain the Hilbert amplitude. Delta and sigma power was thereafter calculated as the square of the absolute Hilbert amplitude. Time windows where tones were played as a result of our auditory stimulation protocol were excluded from the analysis reported in Figs. 1–3. This exclusion minimized the sensory-evoked changes in arousal levels in the analyses. Although the tones were played for a short time compared to the total NREM sleep, and sleep arousals during the stimulation were rare and similar between sham and verum periods, the possibility that auditory stimulation affected arousal levels beyond the stimulation periods cannot be ruled out.

To estimate the spectral slope of the signal, the Fpz signal was first 50 Hz notch filtered, then 0.3 Hz high-pass filtered, and finally 60 Hz low-pass filtered using a Hamming windowed sinc FIR filter from the EEGLAB toolbox function *pop_eegfiltnew*. The spectral power from 0.5 Hz to 45 Hz was estimated using a multitaper approach[81] in the 30 s scored epochs which were subsequently used to extract the slope of a

fitted model using the FOOOF[82] algorithm in the 30 to 45 Hz range, according to Lendner and colleagues[33]. The slope value indicates the slope of the power spectrum in the log-log scale. FOOOF settings were kept default with the exception of minimum peak width limits set to 1 and a maximum number of detectable peaks set to 4. Notably, we identified artifacts as 30 s windows with power values exceeding 3 dB within the frequency range of 0.5 to 45 Hz, resulting in the exclusion of 5.45% of the windows.

To get metrics for sleep micro-architecture dynamics (K-complexes, sleep spindles, and sleep arousals), K-complexes present in the Fpz electrode were visually inspected and manually labeled using Visbrain[79] according to the AASM definition of a K-complex[17]. Due to the inherent difficulty of differentiating K-complexes from increasing background slow wave activity within N3, only K-complexes occurring during N2 and without neighboring slow waves were considered valid. The negative peak of each K-complex was located using MATLAB's *findpeaks* function. As the short initial positive peak seen in K-complexes is not always present, the onset of the K-complex was defined as 550 ms prior to the negative peak, as this is typically the time taken for the sharp negative delineation to occur[46,64]. Fast spindles (12 to 16 Hz) during N2 were automatically detected using the A7 detector from Lacourse et al.[83,84]. The spindle detection performance was visually inspected for each participant, and the sigma correlation parameter for the A7 detector was adjusted accordingly. Spindles lasting less than 300 ms or longer than 2 s were excluded. Spindle clustering in 'trains' was subsequently classified following the Boutin et Doyon[49] criteria: at least 2 consecutive spindles interspaced by at most 6 s. Sleep arousals were automatically detected using a sleep arousal detector algorithm from Fernández-Varela and Alvarez-Estevez[85,86]. The optional "hypnogram" variable was used in the detector algorithm to ensure all detected sleep arousals followed the AASM guidelines[17]. To compare each metric, we excluded epochs where auditory stimuli were presented and normalized pupil size as well as HR to the 5 s before each event (Fig. 3).

A cycle-by-cycle approach was applied for the detection of individual sigma power ISFs. First, the Cz sigma power was low-pass filtered at 0.04 Hz using a 5th-order Butterworth filter. This cut-off frequency was selected as ISFs have been reported to occur below 0.04 Hz in humans[20,21]. Using the MATLAB *findpeaks* function, peaks and troughs were detected. The following criteria had to be met to consider a trough-peak-trough event an ISF: the middle peak and the two troughs occurred during uninterrupted N2; peaks and troughs were at least 15 s apart; troughs were within 100 s from each other; at least 5% of the fluctuation length contained spindles; no auditory stimulation was presented during the cycle; and no artifacts were present. The ISF analysis depicted in Fig. 2 consisted of renormalizing the time scale of sigma power, pupil size, and HR from the time duration of the ISF to the ISF cycle length in terms of percentage cycle. Pupil size and HR were normalized to the average magnitude during each ISF.

### Offline pupil analysis

Video data from the eye tracker goggles had a varying sampling rate higher than 50 Hz. To calculate pupil size, the eye tracker video was first resampled to 50 Hz, and processed using Python 3.8 and DeepLabCut 2.2[31,32]. 36 markers were labeled in each frame as shown in Fig. 1B: 12 marking the palpebral fissure circumference, 12 marking the iris circumference, and 12 marking the pupil circumference. Processing of the markers was conducted in MATLAB. First, an ellipse was fitted to the pupil markers. Then, a line was fitted from each iris marker (I) to the centroid of the ellipse (C). The intersection point of the fitted line with the ellipse (P) was then extracted. Pupil size was calculated as the ratio of the pupil radius to the iris radius using Eq. (1), where N is the number of valid iris markers, $DIC_i$ is the distance from the $i^{th}$ iris marker to the centroid of the ellipse, and $DPC_i$ is the distance from the $i^{th}$

intersection point to the centroid of the ellipse (Fig. 1B).

$$Pupil\ Size = \frac{\sum_i^N \frac{DPC_i}{DIC_i}}{N} \qquad (1)$$

Pupil gaze was calculated by re-referencing the centroid of the pupil markers to the centroid of a polygon generated from the palpebral fissure markers. To account for palpebral fissure twitches (e.g., blink attempts during wake), the palpebral fissure markers were smoothed using a trailing average 40 s moving window.

Right eye pupil size was systematically preprocessed using the guidelines and standardized open-source pipeline published by Kret and Sjak-Shie[87]. Invalid pupil diameter samples representing dilation speed outliers and large deviation from trend line pupil size (repeated four times in a multipass approach) were removed using a median absolute deviation (MAD; multiplier in preprocessing set to 12[87])—an outlier resilient data dispersion metric. Further, temporally isolated samples with a maximum width of 50 ms that border a gap larger than 40 ms were removed (see Supplementary Fig. 13 for a comparison of our pupil assessment to an established eye tracker). Data was then linearly interpolated using the MATLAB function *interp1* to match the 200 Hz sampling rate of polysomnography data and to fill gaps with up to one second of missing data. Relative pupil size was calculated as a percentage change to the mean pupil size of the last 5 s before the events of interest (i.e., spindle trains, K-complexes, sleep arousals, and stimulation windows) or to the average pupil size of the corresponding ISF.

### ECG analysis
ECG R peaks were automatically detected and then visually inspected using the MATLAB-based toolbox PhysioZoo[88]. Data segments consisting of non-detectable peaks or poor quality were excluded from further analyses. HR dynamics is calculated by converting each interval between two R peaks as an inverse of its duration. HR was then linearly interpolated using the MATLAB function *interp1* to match the 200 Hz sampling rate of the polysomnography data. Relative HR was calculated as a percentage change to the mean HR of the last 5 s before the event of interest (i.e., spindle trains, K-complexes, and sleep arousals) or to the average heart rate of the corresponding ISF.

### Auditory stimulation protocol
Within the first 4 h of the complete sleep period, we monitored pupil size and investigated brain responses to administered auditory stimuli depending on arousal levels. For that, we used our custom-developed EEG-feedback controlled stimulation protocol in OpenViBE[76] which was an extension of a previously reported protocol by our group[56,69]. We used three stimulation protocols classified into versions (V): V0 ($N = 4$), V0.5 ($N = 3$), and V1 ($N = 10$). The stimulations took place during NREM sleep which was identified online using prerequisites established in our previous research[56]: no strong anti-correlation for EOG activity, low beta activity, and a high ratio between frontal delta activity to beta activity. The stimulation window start times were marked by the stimulation-algorithm independent of whether sham or verum followed. They only differed in terms of whether real tones were played or not. As our stimulation condition (verum), we used the 45 dB stimulation protocol ISI1 from Huwiler et al.[56], consisting of a 50 ms burst of pink noise at a 45 dB sound level. This burst was presented 10 times in 1 s intervals (ON window) followed by a period of silence (OFF window) where no auditory stimulation was presented. This OFF window was 10 s long in V0 of the stimulation protocol and 20 s long in V0.5 and V1. In V0 of the stimulation protocol, the onset of the ON window was conditional on the detection of the ascending phase of a slow wave during NREM sleep, using a first-order phase-locked loop on the 0.1–40 Hz bandpass filtered signal from electrode Fpz[56,89]. In V0.5 and V1, the onset of the ON window was conditional on the detection

of an increase or a decrease in sigma power during NREM sleep inspired by Cardis et al.[44] This consisted of estimating sigma power using the square of the 12–15 Hz bandpass filtered signal from electrode Cz and comparing current sigma power with the previous 28 s. The thresholds used in V1 for detecting sigma power increases and decreases were optimized in V0.5. A sham stimulation protocol, where no auditory stimulation was presented during ON and OFF windows, was also applied in V0.5 and V1. See Supplementary Table 8 for an overview of what participants were included in each stimulation protocol and the amount of sleep arousals that co-occurred with the ON windows. Auditory stimulation does indeed elevate arousal levels, as reflected by a significant increase in pupil size with tones played when compared to sham (Fig. 4A, D). We therefore excluded ON windows where tones were played from Figs. 1-3 and their corresponding statistical analysis.

The analysis in Fig. 4A–C included the participants from the V1 protocol. Out of the 10 participants in V1, two participants were excluded in Fig. 4A and D due to having less than 3 stimulations with valid pupil size measurements. This led to a final $n = 8$ for Fig. 4A and D and $n = 10$ for Figures B, C, E, and F. To compare the differential effects of high and low arousal levels on responses to auditory stimuli (Fig. 5A, B), we calculated the average pupil size 5 s prior to each stimulation window and used it as a baseline (see offline pupil size analysis). Participants with at least 8 stimulation windows with valid pupil data during the 5 s baseline were pooled together from V0 ($N = 2$) and V1 ($N = 5$) protocols. Then, we separated each participant into the largest and smallest 25% baseline pupil size ranges for each stimulation condition separately.

### Statistics
Statistical analyses were conducted in R (v 3.6.3; R Core Team, Vienna, Austria). Using the R packages lme4[90] and lmertest[91], we computed linear mixed-effects models with several outcome variables (average pupil size, average HR, spectral slope, delta power, or sigma power) and fixed factors with participants as random factors. The fixed factors were sleep stage (Fig. 1F–H), ISF quadrant (Fig. 2G–I), stimulation condition (Fig. 4A–C), or pupil size range and timing of stimulation (Fig. 4D, E). If the one-factor linear mixed-effects models were significant for the fixed effect, we derived post-hoc $p$-values using Satterthwaite's method from the R package lmertest and corrected for multiple comparisons with the Hochberg method using the R package emmeans[92]. If the two-factor linear mixed-effect models were significant for the interaction effect (Fig. 5A, B), we derived post-hoc $p$-values for the contrasts of interest using Satterthwaite's method from the R package lmertest and corrected for multiple comparisons with the Hochberg method. Visual inspection of the residual plots of the linear models did not reveal any obvious deviations from normality or homoscedasticity. Pupil size (Fig. 3A–C) and HR (Fig. 3D–F) were compared to their respective average values 5 s prior to K-complexes, sleep arousals, and spindle trains using paired t-tests corrected for multiple comparisons using Bejamin-Hochberg correction. To statistically compare verum versus sham conditions (Fig. 4A–C), we averaged the data within three time bins (i) Stim$_{early}$ (first 5 s of the auditory stimulation), (ii) Stim$_{late}$ (last 5 s of the auditory stimulation), (iii) Stim$_{post}$ (5 s bin after auditory stimulation was switched off) and applied paired t-tests between verum and sham for each time bin and were corrected for multiple comparisons with the Benjamini-Hochberg method. To calculate inter-participant correlations of pupil size and spectral slope in 30 s epochs across sleep stages, repeated measures correlation analyses were conducted using the R package rmcorr[93]. For the cross-correlation between pupil, HR, and sigma power during ISFs, each signal was z-scored within their ISF and then cross-correlated using the MATLAB function *xcorr*. $p$-values < 0.05 were considered significant. Plots were generated using the R package ggplot2[94] and MATLAB. Where applicable, percentage values

reported in the main body of text are in the format mean ± standard error of the mean.

### Reporting summary

Further information on research design is available in the Nature Portfolio Reporting Summary linked to this article.

## Data availability

Source data for the main figures in the manuscript and sample data to calculate pupil size from DeepLabCut markers are openly available in the ETH Zurich Research Collection, reference number https://doi.org/10.3929/ethz-b-000714436. Raw data that support the findings of this study cannot be made publicly available to protect participants' rights according to Swiss human research law. The de-identified individual participant data that underlies the results of this paper can be accessed by investigators who provide proof of relevant ethical approval for the intended analysis and fulfill data protection measures according to Swiss legal requirements. Requests to gain access to the raw data can be sent to the corresponding author.

## Code availability

Customized codes used to create the main figures and algorithms used to calculate pupil size from DeepLabCut markers have been made publicly available on GitHub, reference number https://doi.org/10.5281/zenodo.14755103.

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

## Acknowledgements

All authors thank their trainees, collaborators, and mentors for the inspiring exchanges on the thematic of this article. The authors would like to express their gratefulness to all participants of the study. This study was supported by the Swiss National Science Foundation (PZ00P3_179795 to C.L., 32003B_207719 to N.W) S.N.M. was supported by the Hochschulmedizin Zürich Flagship project STRESS. This work was also supported by ETH Zurich (22-2 ETH-026) and by the National Research Foundation, Prime Minister's Office, Singapore, under its Campus for Research Excellence and Technological Enterprise (CREATE) program (FHT).

## Author contributions

Writing—review and editing: M.C.D., S.H., S.O., T.L.O., G.B., A.L., N.W., S.N.M., C.L. Writing—original draft: M.C.D., S.H., N.W., S.N.M., C.L. Visualization: M.C.D. Data acquisition: M.C.D., S.O., T.L.O. Data analysis: M.C.D., S.H, S.O., T.L.O., A.L, N.W., S.N.M., C.L. Methodology: M.C.D., G.B., N.W., S.N.M., C.L. Conceptualization, project administration: M.C.D., N.W., S.N.M., C.L. Supervision, funding acquisition: N.W., S.N.M., C.L.

## Competing interests

The authors declare no competing interests.
