## [Transparent Peer Review file · Nature Communications]

Pupil size reveals arousal level fluctuations in human sleep

Corresponding Author: Dr Caroline Lustenberger

Version 0:

Reviewer comments:

Reviewer #1

(Remarks to the Author)

The manuscript introduces a new tool into human sleep research to potentially assess Locus coeruleus activity. The bases is that animal studies have indicated that pupil size is related to arousal level, with most evidence pointing toward LC-NA activity. The new tool is pupillometry during sleep, which was successfully applied during the first 4 hours of nocturnal sleep. The relationship of pupil size to sleep stage, heart rate and spectral slope (as an indicator of excitatory/inhibitory activity), as well as in response to acoustic stimuli, and parallel modulation of evoked sigma and delta activity are systematically investigated. Furthermore, the potential effect of arousal level, as measured by pre-stimulus pupil size was reported to affect magnitude of EEG sigma and delta responses to acoustic stimuli. The novel findings on the relationship of pupil size to the above parameters is potentially of high value to basic research and clinical applications. The compact presentation of a vast number of analyses is also very impressive.

I have one major concern and several detailed comments.

The major concern ambiguity regarding the auditory stimulation (cp. Methods):

The auditory stimulation protocol as described in the Methods is very confusing. Did it consist of a 50 ms burst given every 1 sec for a duration of 10 sec alternately with 20 sec of silence? If so, was this 10 sec on, 20 sec off conditional on some EEG event? Or what does Feedback (l. 586) refer to? This may be detailed in the cited references, but should also be mentioned here.

In addition, three different verum (V) approaches of auditory stimulation are mentioned in the methods. Since the auditory stimulation influence pupil size, sigma and delta power, as shown nicely in Fig.4, it is essential to clearly lay out for which subjects, entering which analyses were the different approaches used. It seems that there are 8 subjects with an identical approach. Since the 10 secon-20 sec off auditory stimulation continued during the complete 4 h recording period (?), I wonder how these different approaches may have affected results presented in Figure 1-3, even if data were collected during the 20 sec off period. Was the occurrence of acoustic stimuli ignored for the corresponding analyses? How do you exclude the effect of auditory stimulation on arousal level (cp. e.g. Schade et al 2020, Nature and Science of Sleep). This should be discussed. Furthermore polysomnographic data for Sham and Verum should be presented.

I suggest to make a table in which for each figure (result) stimulation approach(es), and subject numbers are given. Legends must give the number of subjects included in the data.

This should help clarify what precisely is meant by: "We used different approaches to apply tones for each participant." (l.588-589)

Give the information on how Stimulation was defined in the Sham condition (Fig. 4).

Fig. 5: The largest and smallest quartile were used to distinguish small and large pupil sizes. Pre Stimulation sigma power looks rather similar, but delta power seems to be lower for small pupil size Stim-pre. Please test differences between Stim pre for small and large pupil sizes regarding both sigma and delta activity.

Minor

The Supplementary Video mentioned on line 103 was not found.

Please comment on whether information is known about the asymmetry of any pupil response and include this in explaining the choice of the right eye.

Throughout Fig. 1 axis labelling should be at least as large as used to indicate subject numbers in Fig. 1 I,J.

Show the diagram for the positive correlation between pupil size and spectral slope across ($R=0.34$, $p<0.001$)

Add the p-values in Figs 1 I an J. Are these p- values corrected for multiple comparisons?

l.139: omit and

l. 149: should read Fig. 1E,H, not just Fig. 1E

Add in the legend of Suppl. Fig 1A that N2 was analyzed.

Figure 3 C, F: the centre line of the green area is missing.

Fig. 3, Add that the data were adjusted for multiple comparisons as in Fig. 2 (assuming this was the case). This goes for all figures.

Ref 32, Ref 59 are incomplete

l.263: event-free means without a K-complex, yes?

l. 281-282: ... applied ...and a control condition where no stimuli were applied... Reformulate the sentence.

Figure 4: The light blue text (pre, early) of the x-axis is barely legible

l. 283: four time bins;

l. 287: What is t=0 for Sham? Please write this in the main text. It would be very helpful to indicate the time course of the corresponding auditory stimulation protocols in the 3 panels of Fig. 4, since the different approaches given in the Methods section are confusing. I also suggest integrating V1 , V0.5 and V0 into Figure 4.

l. 300-l.304: These data need to be visualized.

l.475, give name, company, country of the tape used

Suppl., Tables 1 & 3, why are NREM1-3 instead of the standard N1-N3 terminology used.

Suppl., Table 7, clarify that "Effect" means 5s activity prior to the corresponding events.

Reviewer #2

(Remarks to the Author)

General assessment of the work

In their manuscript, Carro-Domínguez and colleagues measure pupil size during sleep and relate this to several sleep metrics, assessed using polysomnography, as well as heart rate. In addition, they use an auditory stimulation protocol to investigate how pupil size relates to neural evoked responses.

In a sample of about 17 younger adults, the authors find that pupil size differs across sleep stages, comparable to recent findings in rodents, and correlates with EEG spectral slope. During N2 sleep, sigma power shows slow fluctuations that are related to those of pupil size, but phase shifted. Zooming in on several specific events in the microarchitecture of N2 sleep (k-complexes, spindle clusters, sleep arousals), the authors show that each of these events is followed by relative increases in pupil size. Finally, using rhythmic auditory stimulation, they show event-related pupil dilation during sleep and that evoked EEG responses are associated with baseline pupil size.

This study builds on accumulating evidence associating luminance-independent fluctuations in pupil size to neuromodulation. Moreover, it expands recent investigations in rodents showing a role of the noradrenergic locus coeruleus in regulating sleep rhythms. Taken together, I consider the manuscript timely and of interest to a broader scientific community. The approach of measuring pupil size with eyes open during sleep fills a gap in the literature, although data loss is considerable for some participants and comparability (of sleep metrics) with sleep with eyes closed remains to be established. I provide several suggestions below.

Major comments

1. I think the clarity of the manuscript could be increased by including a definition of what the authors mean by the term "arousal." This may seem trivial, but in my opinion, different definitions for this term exist (e.g., many definitions are possible, for the purposes of this review, the term arousal refers to a continuum of sensitivity to environmental stimuli. [1]). This comment is reinforced by the fact that the authors refer to several different types of arousal throughout the manuscript (e.g., central arousal, cortical arousal, sleep arousal), and it remains unclear if/how these are associated. In this context, it would be instructive if this definition explicitly considered how these arousal types are mechanistically related to neuromodulation.

2. I am aware that it is difficult to track pupil size during sleep – but I also think that it may differ significantly from participants' usual sleep situation. As the authors have collected EEG/ECG with and without eye tracking, it would be good to use also the latter recordings to evaluate if keeping one eye open during sleep is associated with altered sleep rhythms. In addition, I would encourage the authors to report the mood and sleep quality data they collected (~line 495).

3. I found the description of the eye tracking system lacked several key aspects, including for instance calibration/validation procedures, sampling rate, accuracy etc.

Could the authors compare their deep-lab cut-based pupil assessment to more established eye trackers to indicate the validity?

4. Some work in animals suggests that raw pupil size and its first derivative are differentially associated with cholinergic and noradrenergic neuromodulation [2]. The authors could complement their analyses by also including the first derivative of the pupil time series.

5. Figure 5 reports evoked EEG responses in relation to pre-stimulation pupil size. Here the authors select 2 out of 4 bins with low and high pupil size. However, a lot of research in rodents[e.g., 3,4] shows (inverted) u-shaped associations with pupil size, which will not be detected with the current approach. I would therefore suggest using all the data collected, possibly even with a finer binning. More generally, I think the manuscript could be improved by extending their discussion of the relation between pupil-indexed neuromodulation and brain states beyond the sleep field.

Related to this analysis, it would be interesting to also include associations between baseline and evoked pupil dilation (cf.

e.g., [5]).

6. In their discussion, the authors write that locus coeruleus neuromodulation may be involved in several of the reported sleep metrics. (e.g., line~364, 396) I would like to encourage the authors to be more specific and state the mechanisms of *how* neuromodulation shapes sleep rhythms.

Minor comments

1. Introduction: The accessibility of the manuscript could be increased by including brief introductions for the different sleep metrics (e.g., spindles, ~ line 66). For instance, k-complexes are only introduced at the end of the paper (~ line 396) – providing this information earlier would help readers who are not well versed in the sleep field.
2. Results: Since there was considerable eye tracking data loss, the authors should report (can be SI) if data loss was uniformly distributed across the pupil/sleep stages.
3. Results: “Arousal levels can be used to define sleep depth and have been shown to depend on the sleep macro-architecture” – seems like there is a citation missing here.
4. Figure 1 B could be improved by including and labeling the different parameters that were used to estimating pupil size.
5. Results/SI: I could not find the statistical results (p-value) related to the cross-correlations, could these be added, along with marking the significant lags in the figure (incl. a control for multiple comparisons)?
6. Results: “Arousals from sleep as defined by AASM guidelines are brief perturbations to sleep which are characterized by large and sudden changes in the theta, alpha, or beta frequencies in the EEG power spectra, which can be accompanied by EMG activation.” – this section could be clearer when including the directionality of spectral changes.
7. Figure 3 E: While I do not have much experience with heart rate data, I was surprised by the variance difference between Fig 3 E relative to 3D/F. Could the authors elaborate on this?
8. Line 618, repeated word: arousal

Literature

- 1 Berridge, C.W. (2008) Noradrenergic modulation of arousal. *Brain Res. Rev.* 58, 1–17
- 2 Reimer, J. et al. (2016) Pupil fluctuations track rapid changes in adrenergic and cholinergic activity in cortex. *Nat. Commun.* 7, 13289
- 3 McGinley, M.J. et al. (2015) Cortical membrane potential signature of optimal states for sensory signal detection. *Neuron* 87, 179–192
- 4 McGinley, M.J. et al. Waking State: Rapid Variations Modulate Neural and Behavioral Responses. , *Neuron*, 87. 23-Sep- (2015) , Cell Press, 1143–1161
- 5 Collins, L. et al. (2021) Vagus nerve stimulation induces widespread cortical and behavioral activation. *Curr. Biol.* 31, 2088–2098.e3

Reviewer #3

(Remarks to the Author)

Thank you for inviting me to review this manuscript, which used an inventive approach – namely, taping a participant’s eyelid open and then having them fall asleep – to track pupillary dynamics across sleep stages. The authors then compared these recordings to the power within different frequency bands estimated from scalp electrophysiology and also in response to auditory tones. These data provide evidence that the pupil can be used to track responses to stimuli during sleep.

* The authors may wish to compare their results to other studies that have tracked pupil diameter across sleep stages in animals, such as Tsunematsu et al., 2020 eLife.

* Sorry if I missed this, but how were the differences in ambient light levels across sleep and wake controlled for?

* Figure 1H – was there a reason that the authors didn’t standardize the HR patterns across subjects (as they did for pupil size and spectral slope)?

* Figure 1 – rather than plotting a single value for each brain state per subject, I wonder whether it would be more informative to plot the distribution of pupil sizes as a function of brain state. This will help to provide a deeper intuition for the range of pupillary dilation across different brain states. Given that the pupillary traces shown in 1C are distinctly non-linear, it is difficult to know whether the mean value within each state is an effective summary of the dependent variable. The fact that this measure is used to make inferences later in the manuscript makes this an important baseline issue to resolve in order to ensure robust interpretability of the authors work.

* Figure 2 – did the authors account for differences in the spectral content of the signals they were comparing? By eye, the pupils + HR signals are quite low-frequency relative to the sigma power fluctuations, whereas the K-complexes are relatively

high-frequency.

* Figure 3 - The variance in the plots is quite large (as shown by the grey lines, which is excellent), though the use of S.E. rather than S.D. to define the error bars might appear somewhat misleading to the reader.

* L286 – The correct name of the correction used is “Benjamini”, not “Benjamin”

* Did the auditory stimulations during sleep rouse the participants?

* Discussion: The statement “Therefore, we propose that pupil size dynamics might represent a non-invasive way to track arousal levels” disregards a large literature that has already made similar inferences. I suggest re-wording the main conclusion to be more specific to the results of this experiment.

Version 1:

Reviewer comments:

Reviewer #1

(Remarks to the Author)

1. Auditory Stimulation Protocol (Methods):

(Page 2, line 680 in the manuscript)

Issue: The reviewer found the auditory stimulation protocol confusing, specifically about the 10s on and 20s off cycle, and whether this was EEG event-based.

Feedback: The response clarifies the stimulation protocol well. Although citations for the EEG feedback developed you are given, could you please add a sentence or two to make the detection processes could more clear.

2. Participant Count and Protocol Variation:

(Supplementary Table 8 on page 8 of the main text)

Issue: Reviewer wanted clarification on the different stimulation approaches (verum vs. sham) and their effects on Figures 1–3.

Feedback: This concern has been well addressed by providing details on participant numbers in each protocol and generating a new table in the supplement. However, the question regarding the exclusion of stimulation affecting arousal level is not addressed enough. This should be discussed as a potential limitation if it cannot be a priori excluded. Also, since the usage of the terms “sleep arousals” (as determined by PSG) and “arousal levels” does not seem to be stringent throughout the manuscript, the differential (or similar) effect of stimulation on both measures should be more clearly made.

3. Arousal Definition:

(Page 2, line 53 (arousal levels) and page 7, line 253 (sleep arousals))

Issue: Reviewer requested a clear definition of “arousal” and how different types of arousal are related.

Feedback: This has been addressed well, however comprehension on how “arousal level” is defined would benefit greatly from a more concise description. For instance, it seems on the one hand pupil size is used as a proxy for arousal level (p. 8, l. 317), on the other hand, it is assumed that pupil size reflects arousal level, and the effect on EEG activity is the output measure (p. 8, l. 335).

4. Auditory Stimulation and Arousal Levels:

(Page 14, line 601 in the manuscript)

Issue: There was confusion about how stimulation might affect arousal levels.

Feedback: The below definition was given (p. 3, 3rd to last paragraph), I suggest writing “...which in REM sleep must be accompanied by increases in EMG amplitude”

Reviewer #2

(Remarks to the Author)

I thank the authors for their thorough revision, which addressed my concerns.

Below, I provide some minor suggestions that the authors may address:

- Regarding my initial major comment #2 (open vs closed eye sleep differences): I agree that without a proper control night, the provided comparison is not conclusive (perhaps the authors even have access to such data?). To provide a reference for future follow up studies, I would nonetheless suggest including the figure the authors added to their rebuttal in the supplementary information.

- Regarding my initial minor comment #2 (data loss): One participant shows very little valid data (25 % max during W, dropping to values close to zero for the remaining stages), was this participant excluded from analyses? Would be good to clarify this in the legend.

- Regarding the analyses reported in fig 5, this paper showing state-dependent effects of locus coeruleus stimulation on cortical electrophysiology in anesthetized rodents could be useful: <https://pubmed.ncbi.nlm.nih.gov/29093170/>

- I enjoyed reading a recent related publication, demonstrating closed-eye pupil tracking using an alternative method that the authors could qualitatively compare to their method:

Reviewer #3

(Remarks to the Author)

The authors have adequately addressed my concerns.

Version 2:

Reviewer comments:

Reviewer #1

(Remarks to the Author)

The authors have addressed all my concerns nicely. I just have a few minor comments:

l.691-692: I believe there is a typo in the unit of this sentence:

"...level. This burst was presented every second for a duration of 10 seconds (ON window) followed by a period of silence (OFF window) where no auditory stimulation was presented...."

l.299 'N2 that were tones were', omit 'that'

l. 300, the term "general N2 levels" is first used here. Since this description 'slips in' as a frequently used description, but is not clearly defined, I suggest to add in brackets here e.g., (i.e., pupil size during N2, excluding the 5 s before and 5 s after an auditory stimulation).

Discussion

Although such analyses would go beyond the scope of this manuscript, in the discussion (around l.423 ff) it should be addressed that other parameters derived from the ECG, and thus reflecting autonomic nervous system activity may reveal a/another relationship to sleep stages/ISF and pupillometry (e.g., Naji et al. 2019, *Neurobiology of Learning and Memory*).

Reviewer #2

(Remarks to the Author)

The authors have adequately addressed my concerns and I congratulate them on this nice manuscript!

We would like to thank the editor and the reviewers for taking the time to assess our work. We thoroughly analyzed the reviewers' comments and modified parts of the manuscript based on their feedback. Below, we provide a point-by-point response to each comment (in blue). We believe that addressing the comments has strengthened our manuscript and sincerely appreciate the feedback we received.

Reviewer #1 (Remarks to the Author):

The manuscript introduces a new tool into human sleep research to potentially assess Locus coeruleus activity. The basis is that animal studies have indicated that pupil size is related to arousal level, with most evidence pointing toward LC-NA activity. The new tool is pupillometry during sleep, which was successfully applied during the first 4 hours of nocturnal sleep. The relationship of pupil size to sleep stage, heart rate and spectral slope (as an indicator of excitatory/inhibitory activity), as well as in response to acoustic stimuli, and parallel modulation of evoked sigma and delta activity are systematically investigated. Furthermore, the potential effect of arousal level, as measured by pre-stimulus pupil size was reported to affect magnitude of EEG sigma and delta responses to acoustic stimuli. The novel findings on the relationship of pupil size to the above parameters is potentially of high value to basic research and clinical applications. The compact presentation of a vast number of analyses is also very impressive.

We appreciate the reviewer's constructive feedback and are happy to hear that they were impressed with the analysis. In response to the reviewer's comments, we have conducted additional analyses and clarified some critical points that were raised. We thank the reviewer for their comments.

Major Comments

The major concern ambiguity regarding the auditory stimulation (cp. Methods):

The auditory stimulation protocol as described in the Methods is very confusing. Did it consist of a 50 ms burst given every 1 sec for a duration of 10 sec alternately with 20 sec of silence? If so, was this 10 sec on, 20 sec off conditional on some EEG event? Or what does Feedback (l. 586) refer to? This may be detailed in the cited references, but should also be mentioned here.

In addition, three different verum (V) approaches of auditory stimulation are mentioned in the methods. Since the auditory stimulation influence pupil size, sigma and delta power, as shown nicely in Fig.4, it is essential to clearly lay out for which subjects, entering which analyses were the different approaches used. It seems that there are 8 subjects with an identical approach. Since the 10 sec on-20 sec off auditory stimulation continued during the complete 4 h recording period (?), I wonder how these different approaches may have affected results presented in Figure 1-3, even if data were collected during the 20 sec off period. Was the occurrence of acoustic stimuli ignored for the corresponding analyses? How do you exclude the effect of auditory stimulation on arousal level (cp. e.g. Schade et al 2020, Nature and Science of Sleep). This should be discussed. Furthermore polysomnographic data for Sham and Verum should be presented.

I suggest to make a table in which for each figure (result) stimulation approach(es), and subject numbers are given. Legends must give the number of subjects included in the data.

This should help clarify what precisely is meant by: "We used different approaches to apply tones for each participant." (l.588-589)

Give the information on how Stimulation was defined in the Sham condition (Fig. 4).

We thank the reviewer for this valuable comment. We have now adjusted the first paragraph of the stimulation protocol in the methods section for clarification (page 16 line 680) as follows: *Within the first four hours of the complete sleep period, we monitored pupil size and investigated brain responses to administered auditory stimuli depending on arousal levels. For that, we used our custom-developed EEG-feedback controlled stimulation protocol in OpenViBE⁷³ which was an extension of a previously reported protocol by our group^{55,67}. We used three stimulation protocols classified into versions (V): V0 (N=4), V0.5 (N=3), and V1 (N=10). The stimulations took place during NREM sleep which was identified online using prerequisites established in our previous research⁵⁵: no strong anti-correlation for EOG activity, low beta activity, and a high ratio between frontal delta activity to beta activity. The stimulation window start times were marked by the stimulation-algorithm independent whether sham or verum followed. They only differed in terms of whether real tones were played or not. As our stimulation condition (verum), we used the 45dB stimulation protocol ISI1 from Huwiler et al.⁵⁵, consisting of a 50ms burst of pink noise at a 45dB sound level. This burst was presented every second for a duration of 10 seconds (ON window) followed by a period of silence (OFF window) where no auditory stimulation was presented. This OFF window was 10s long in V0 of the stimulation protocol and 20s long in V0.5 and V1. In V0 of the stimulation protocol, the onset of the ON window was conditional on the detection of the ascending phase of a slow wave during NREM sleep. In V0.5 and V1, the onset of the ON window was conditional on the detection of an increase or a decrease in sigma power during NREM sleep inspired by Cardis et al.⁴⁴ The thresholds used in V1 for detecting sigma power increases and decreases were optimized in V0.5. A sham stimulation protocol, where no auditory stimulation was presented during ON and OFF windows, was also applied in V0.5 and V1. See Supplementary Table 8 for an overview of what participants were included in each stimulation protocol and the amount of arousals that co-occurred with the ON windows. Auditory stimulation does indeed elevate arousal levels, as reflected by a significant increase in pupil size with tones played when compared to sham (Figure 4A,D). We therefore excluded ON windows where tones were played from Figures 1, 2, and 3 and their corresponding statistical analysis.*

We have created a table (Supplementary Table 8; see below) laying out which participants had what stimulation approach and in what stimulation analysis (Figures 4 and 5) they were included in. This table also contains the number of arousals, as defined by the AASM guidelines that took place during the ON windows of each stimulation protocol. We hope that this table clarifies that our stimulation protocol did not increase the likelihood of evoking arousals, as defined by the AASM guidelines (abrupt changes in EEG frequency, suggestive of an awake state, which must be accompanied by increases in EMG amplitude during REM).

Participant number	Stimulation protocol	Stimulation analysis included in	Number of sham windows	Number of verum windows	Number of arousals during sham windows	Number of arousals during verum windows	Sham windows with arousals (%)	Verum windows with arousals (%)
1	V0	None	-	0	-	0	-	0
2	V0	Figure 5 All	-	303	-	4	-	0.33
3	V0	None	-	85	-	3	-	1.17
4	V0	Figure 5 All	-	94	-	1	-	1.06
5	V0.5	None	12	14	0	0	0	0
6	V0.5	None	0	0	0	0	NaN	NaN
7	V0.5	None	1	1	0	0	0	0
8	V1	Figure 4 All and Figure 5 All	74	72	1	1	1.35	1.39
9	V1	Figure 4 All	15	14	0	0	0	0
10	V1	Figure 4 All and Figure 5 All	66	64	0	0	0	0
11	V1	Figure 4 All and Figure 5 All	46	45	0	0	0	0
12	V1	Figure 4 All	56	55	0	1	0	1.82
13	V1	Figure 4 All and Figure 5 All	48	44	0	0	0	0
14	V1	Figure 4 All and Figure 5 All	47	45	0	0	0	0
15	V1	Figure 4 All	55	54	0	0	0	0
16	V1	Figure 4 B,C,E,F	15	15	1	0	6.67	0
17	V1	Figure 4 B,C,E,F	8	7	0	0	0	0

Supplementary Table 8. Stimulation protocols. Description of what participants were included in each stimulation protocol and corresponding analysis in Figure 4 and Figure 5. The number of sham and verum windows and the number of arousals co-occurring with these windows are reported.

We also clarified that ON windows where tones were played were excluded from Figures 1, 2, and 3 and their corresponding statistical analysis on page 14 line 601 as follows: *Time windows where tones were played as a result of our auditory stimulation protocol were excluded from the analysis reported in Figures 1-3.*

We have now clarified the definitions of arousals on page 7 line 253 (as suggested by reviewer #2, see below) and arousal levels on page 2 line 53: *According to these criteria, sleep arousals are characterized by abrupt changes in EEG frequency, suggestive of an awake state, which must be accompanied by increases in EMG amplitude during REM^{4,47}.*

Although the amount of arousals defined according to the AASM criteria remained largely unaffected by our auditory stimulation (as reported in Supplementary Table 8), pupil dilation associated with auditory stimulation does suggest that arousal levels were affected by the stimulation.

Additionally, each participant's corresponding gray line in Figure 5 is now clearly labelled. The Figure 5 legend further indicates which participant was involved in what stimulation protocol.

Figure 5. EEG responses to auditory stimulation at small and large pupil size baselines. (A) Sigma power 5s before ($Stim_{pre}$) and during the first 5s ($Stim_{early}$) of verum stimulation of trials with large (highest quartile) and small pupil baselines (lowest quartile) to group into high and low arousal level trials. We found an interaction effect indicating increased sigma power from $Stim_{pre}$ to $Stim_{early}$ in the small pupil baseline condition ($5.53 \pm 2.04 \mu V^2$) but not in the large pupil baseline condition ($0.08 \pm 2.17 \mu V^2$), $F_{\text{sigma} \times \text{pupilrange}}(1, 21) = 5.66$, $p = 0.027$, post-hoc comparison $Stim_{pre}$ to $Stim_{early}$ with small pupil baseline: $F(1, 7) = 8.60$, $p = 0.044$, post-hoc comparison $Stim_{pre}$ to $Stim_{early}$ with large pupil baseline: $F(1, 7) = 0.002$, $p = 0.97$. **(B)** Delta power during $Stim_{pre}$ and during $Stim_{early}$ of verum stimulation of trials with large (highest quartile) and small pupil baselines (lowest quartile) to group into high and low arousal level trials. We found an interaction effect indicating increased delta power from $Stim_{pre}$ to $Stim_{early}$ in the small pupil baseline condition but not in the large pupil baseline condition $F_{\text{delta} \times \text{pupilrange}}(1, 28) = 6.77$, $p = 0.015$, post-hoc comparison $Stim_{pre}$ to $Stim_{early}$ with small pupil baseline: $F(1, 14) = 7.58$, $p = 0.031$, post-hoc comparison $Stim_{pre}$ to $Stim_{early}$ with large pupil baseline: $F(1, 14) = 0.25$, $p = 0.623$. The numbers next to the gray lines correspond to the participant number. The stimulation protocol V0 was used in 2 and 4, and V1 for the rest. The horizontal black lines connecting pupil size ranges are the interaction effects between pupil size range and $Stim_{pre}$ - $Stim_{early}$ power values. p-values are based on post-hoc t-test and are adjusted for multiple comparisons using Benjamini-Hochberg correction.

Lastly, we have now added Supplementary Figure 8 showing polysomnographic data in the form of EEG, EOG vertical eye movements, and EMG traces at one exemplary trial in verum and sham for each participant that was included in all analysis reported in Figure 4 and in Figure 5, as referenced in Supplementary Table 8. This Supplementary Figure is now referenced in the main text on page 8 line 321.

Supplementary Figure 8. Polysomnography data during stimulation windows. Example of polysomnography data during stimulation windows for both verum and sham trials in participants of stimulation protocol V1 that were included in the analysis reported in Figure 4, 5 in the main manuscript (see Supplementary Table 8 for details). Each blue quadrant contains an exemplary trial from the verum and sham from each participant. Data includes EEG, eye movements in vertical direction derived from EOG (EOG vertical), and EMG. EEG, EOG, and EMG signals were notch filtered at 50Hz and bandpass filtered in the 0.5 to 35Hz, 0.5 to 35Hz, and 10 to 90Hz frequency range (respectively) using MATLAB 2nd order Butterworth filters. Time zero is the detection of the stimulation window.

Fig. 5: The largest and smallest quartile were used to distinguish small and large pupil sizes. Pre Stimulation sigma power looks rather similar, but delta power seems to be lower for small pupil size Stim-pre. Please test differences between Stim pre for small and large pupil sizes regarding both sigma and delta activity.

We thank the reviewer for suggesting this important analysis. Power values were lower for small pupil size stim-pre than for large pupil size for both delta ($F(1,7)=8.97$, $p=0.020$), and sigma ($F(1,14)=7.40$, $p=0.017$). Based on Figure 2 and 3 a close interdependence of pupil with sigma power bursts (in the form of the infraslow fluctuation and spindle trains) and delta power bursts (in the form of K-complexes likelihood and isolated K-complexes) has been demonstrated. Pupil size is lowest right before the onset of these bursts and therefore strong baseline differences in pupil are expected to be present in power differences in sigma and delta. Statistically, it is not possible to separate this dependence as confirmed by consultation with a statistical expert from ETH Zurich. Therefore, we did not include this analysis in the manuscript.

Minor

The Supplementary Video mentioned on line 103 was not found.

We sincerely apologize for not including the video in the original submission. It has now been added.

Please comment on whether information is known about the asymmetry of any pupil response and include this in explaining the choice of the right eye.

Pupil size asymmetry does exist (anisocoria) but was not explicitly considered here. We refrained from tracking both eyes to reduce risks related to keeping the eye open that were pointed out by the ethics committee. We chose the right eye to have a standardized procedure for one eye and because it is commonly the dominant eye for visual input. However, we did not check for eye dominance in this study.

Throughout Fig. 1 axis labelling should be at least as large as used to indicate subject numbers in Fig. 1 I,J. We corrected the axis labelling in Figure 1.

Show the diagram for the positive correlation between pupil size and spectral slope across ($R=0.34$, $p<0.001$).

We have created a plot that shows the for each participant repeated measures correlation of pupil size and spectral across all sleep stages. This has now been added in the form of Supplementary Figure 3. This figure supports the significant effect sizes of the repeated measures correlation we report as each participant depicts a similar correlation. This figure is now referenced in the main text in page 4 line 168 as follows: *Repeated measures correlation analyses revealed a significant positive correlation between pupil size and spectral slope across ($R=0.34$, $p<0.001$, see Supplementary Fig. 3 showing this within-individual association) [...]*

Supplementary Figure 3. Pupil size versus spectral slope for all sleep stages. Repeated measures correlation of pupil size versus spectral slope for all sleep stages: $R=0.34$, $p<0.001$, $N=17$. Different colors represent different participants.

Add the p-values in Figs 1 I and J. Are these p-values corrected for multiple comparisons?

We thank the reviewer for this constructive feedback. We have now added the p-values (corrected for multiple comparisons using Benjamini-Hochberg correction) in Figure I and J in addition to editing the corresponding references to these p-values in the main text (page 4 lines 170 and 176).

l.139: omit and **Omitted**.

l. 149: should read Fig. 1E,H, not just Fig. 1E. **Corrected**.

Add in the legend of Suppl. Fig 1A that N2 was analyzed. **Added and renamed to Suppl. Fig. 4A**

Figure 3 C, F: the centre line of the green area is missing. **Added**.

Fig. 3, Add that the data were adjusted for multiple comparisons as in Fig. 2 (assuming this was the case). This goes for all figures. **Added to all figure legends where applicable**.

Ref 32, Ref 59 are incomplete. **Completed**.

l.263: event-free means without a K-complex, yes?

Event-free was referring to moments where tones were not played. We have now clarified this in the paper on page 7 line 299 as follows: *To shed light on the pupil size and HR dynamics surrounding K-complexes, arousals, and spindles, we compared the baseline of each event to the*

average pupil size ($F(3, 39.30)=3.68, p=0.020$) and HR ($F(3, 40.03)=6.34, p=0.001$) during periods of N2 that were tones were not played.

l. 281-282: ... applied ...and a control condition where no stimuli were applied... Reformulate the sentence. Reformulated.

Figure 4: The light blue text (pre, early) of the x-axis is barely legible. The text has been made more visible by using a dark blue color.

l. 283: four time bins. We compared sham vs verum at three time bins $Stim_{early}$, $Stim_{late}$, $Stim_{post}$. These three time bins were normalized to $Stim_{pre}$. We understand how the original formulation was confusing so we have clarified this sentence on page 8 line 322 by stating that the data was normalized to $Stim_{pre}$ and subsequently kept the original formulation. This now reads: *To statistically compare verum versus sham conditions, we normalized the data to $Stim_{pre}$ and averaged within three time bins [...]*

l. 287: What is $t=0$ for Sham? Please write this in the main text. It would be very helpful to indicate the time course of the corresponding auditory stimulation protocols in the 3 panels of Fig. 4, since the different approaches given in the Methods section are confusing. I also suggest integrating V1, V0.5 and V0 into Figure 4.

Time zero is the start of the stimulation window. This is now mentioned in page 9 line 364: *Time zero is the start of the stimulation window.* The stimulation window start times were marked by the stimulation-algorithm independent whether sham or verum followed. They only differ in terms of whether real tones were played or not.

However, we do understand that the auditory stimulation protocols were not fully clear. We believe that the new version of the auditory stimulation section in the methods on page 16 line 680 provides sufficient detail and clarity for the reader to know what stimulation protocols were used in Figure 4. Furthermore, we now refer to the new Supplementary Table 8 in the legend of Figure 4 as this table provides information about the number of participants from each stimulation protocol that were used in this analysis.

l. 300-l.304: These data need to be visualized. A visualization conveying the referenced results can now be seen as Supplementary Figure 9 and referenced on page 9 line 344.

Supplementary Figure 9. Pupil size prior to stimulation versus general N2 levels. Pupil size during N2 compared to large pupil size at baseline prior to stimulation and small pupil size at baseline prior to stimulation.

l.475, give name, company, country of the tape used. Added on page 13 line 567.

Suppl., Tables 1 & 3, why are NREM1-3 instead of the standard N1-N3 terminology used. We thank the reviewer for pointing out this inconsistency. The tables have been adjusted to match the N1-N3 terminology.

Suppl., Table 7, clarify that “Effect” means 5s activity prior to the corresponding events. The term effect was indeed unclear and has been replaced with the proposed terminology by the reviewer.

Reviewer #2 (Remarks to the Author):

General assessment of the work

In their manuscript, Carro-Domínguez and colleagues measure pupil size during sleep and relate this to several sleep metrics, assessed using polysomnography, as well as heart rate. In addition, they use an auditory stimulation protocol to investigate how pupil size relates to neural evoked responses.

In a sample of about 17 younger adults, the authors find that pupil size differs across sleep stages, comparable to recent findings in rodents, and correlates with EEG spectral slope. During N2 sleep, sigma power shows slow fluctuations that are related to those of pupil size, but phase shifted. Zooming in on several specific events in the microarchitecture of N2 sleep (k-complexes, spindle clusters, sleep arousals), the authors show that each of these events is followed by relative increases in pupil size. Finally, using rhythmic auditory stimulation, they show event-related pupil dilation during sleep and that evoked EEG responses are associated with baseline pupil size.

This study builds on accumulating evidence associating luminance-independent fluctuations in pupil size to neuromodulation. Moreover, it expands recent investigations in rodents showing a role of the noradrenergic locus coeruleus in regulating sleep rhythms. Taken together, I consider the manuscript timely and of interest to a broader scientific community. The approach of measuring pupil size with eyes open during sleep fills a gap in the literature, although data loss is considerable for some participants and comparability (of sleep metrics) with sleep with eyes closed remains to be established. I provide several suggestions below.

We are thrilled that the reviewer viewed our study as timely and of interest to a broader scientific community, with compelling findings. Their critical comments were invaluable, prompting further analyses and significant revisions throughout the manuscript.

Major comments

1. I think the clarity of the manuscript could be increased by including a definition of what the authors mean by the term "arousal." This may seem trivial, but in my opinion, different definitions for this term exist (e.g., many definitions are possible, for the purposes of this review, the term arousal refers to a continuum of sensitivity to environmental stimuli. [1]). This comment is reinforced by the fact that the authors refer to several different types of arousal throughout the manuscript (e.g., central arousal, cortical arousal, sleep arousal), and it remains unclear if/how these are associated. In this context, it would be instructive if this definition explicitly considered how these arousal types are mechanistically related to neuromodulation.

We thank the reviewer for raising this important point and we now use only 'arousal levels' and 'sleep arousals'. To increase the clarity of the manuscript, we have made several changes regarding the terminology used in this paper to describe arousal:

1. We now reference two types of arousal: arousal levels and sleep arousals.
 - a. "Arousal levels" are now defined on page 2 line 53 as follows: *Arousal can be viewed as a continuum of engagement a system has with incoming stimuli, with different levels at cortical regions defining vigilance states. Within the lower states of vigilance, collectively termed sleep, arousal level dynamics can even vary significantly, with substantial consequences for our overall sleep quality. They can be subtle and temporary, detected primarily by neurophysiological markers such*

as K-complexes^{1,2}, spindles³, or very short bouts of wakefulness, defined as sleep arousals⁴. Alternatively, arousal level dynamics can be large and sustained, resulting in transitions from low vigilance states of sleep, such as non-rapid eye movement (NREM) sleep, to larger states manifesting as wakefulness, or from low to even lower states manifesting as rapid eye movement (REM) sleep. The likelihood of changing arousal levels is primarily dependent on the sensitivity an arousal-related system has in processing incoming information, and the saliency of the information. Central drivers of arousal levels include nuclei that also modulate pupil size such as the noradrenergic (NA) locus coeruleus (LC)^{5,6}, the cholinergic nucleus basalis of Meynert^{7,8}, serotonergic neurons in the dorsal raphe^{9,10}, and orexin/hypocretin-producing neurons in the lateral hypothalamus^{11,12}. Other drivers of arousal that project to these nuclei include neuropeptide S¹³, the cortex¹⁴, and the autonomic nervous system¹⁵. The activity of these interconnected systems subsequently modulates arousal levels at cortical regions characterizing sleep as a state of low arousal level and disconnection from the environment with reduced responsiveness to external stimuli.

- b. "Sleep arousals" are now defined on page 7 line 253 as follows: *According to these criteria, sleep arousals are characterized by abrupt changes in EEG frequency, suggestive of an awake state, which must be accompanied by increases in EMG amplitude during REM.* In essence, sleep arousals capture snapshots of brief arousal level changes that are elevated enough to manifest as EEG-derived cortical activity.
2. We replaced references to central and cortical arousal on page 4 line 147. This now reads: *Arousal levels can be used to define sleep depth and have been shown to depend on the sleep macro-architecture^{33,34}. To test this basic principle, we determined whether (i) pupil size (Fig. 1C), an established indicator of activity from arousal-regulating systems in brainstem and basal forebrain regions during wakefulness^{8,22,35} (ii) EEG spectral slope (Fig. 1D), a marker of cortical activity modulated by the aforementioned arousal-regulating systems³³, and (iii) HR (Fig. 1D), a marker for cardiovascular activation, differed across sleep stages using linear mixed effect models and calculating post-hoc t-tests corrected for multiple comparisons using Benjamini-Hochberg correction.*

2. I am aware that it is difficult to track pupil size during sleep – but I also think that it may differ significantly from participants' usual sleep situation. As the authors have collected EEG/ECG with and without eye tracking, it would be good to use also the latter recordings to evaluate if keeping one eye open during sleep is associated with altered sleep rhythms. In addition, I would encourage the authors to report the mood and sleep quality data they collected (~line 495).

We thank the reviewer for this valuable input. As first sanity check, we calculated the percentage of time spent in each sleep stage in the sleep window when the right eye is taped open (Eye Open, first part of the night) versus in the sleep window after the tape is removed and the eye can be closed (Eye Closed, second part of the night; see figure below). When the eye is taped open, individuals stay more time in wake (8.36±3.70%, p=0.048), N1 (5.37±2.16%, p=0.041), and in N3 (7.23±2.21%, p=0.021) when compared to time spent with eyes closed and less time in REM (-14.63±1.99%, p<0.001) whereas time spent in N2 (-2.25±3.76%, p=0.558) was not significantly different. It is important to keep in mind that healthy sleep is characterized by changes in macrostructure throughout the night. In the early hours of sleep, wake and N1 bouts are typically longer. Also, in the early hours of sleep N3 bouts are longer than REM bouts, whereas in the second half of the night, REM bouts are longer than N3 bouts. As such, our additional analysis only revealed expected changes in sleep macrostructure and we found no clear evidence that sleeping with the eyes open caused major sleep disturbances. However, to truly answer this question, one would have to run a larger study with multiple sleep sessions in the laboratory and in addition compare sleep macrostructure when the eye is taped open in the second half of the night instead of the first half as done here.

Sleep macrostructure with right eye taped open vs right eye closed. Percentage of time spent in each sleep stage in the sleep window when the right eye is taped open (Eye Open) versus in the sleep window after the tape is removed and

the eye can be closed (Eye Closed). N1: NREM stage 1; N2: NREM stage 2; N3: NREM stage 3; R: rapid eye movement (REM). The horizontal black lines connecting pupil size ranges are the interaction effects between pupil size range and Stimpre-Stimearly power values. *p*-values are based on post-hoc *t*-test and are adjusted for multiple comparisons using Benjamini-Hochberg correction.

Furthermore, we now report the mood data from the Vigor Affect questionnaire and the Karolinska Sleepiness Scale in Supplementary Figure 12A-B. These results indicate that participants had similar mood levels, both positive and negative, in the evening and morning, with an overnight reduction in sleepiness comparable to values from other studies (Åkerstedt 2017). The Profile of Mood States questionnaire was also asked. However, this questionnaire is not reported as it evaluates mood over the past 2 weeks, rather than the current emotional state. The responses to the sleep quality questions are summarized and explained in Supplementary Figure 12C. This supplementary figure is cited in the main text on page 14 line 587.

Åkerstedt, T., Hallvig, D., & Kecklund, G. (2017). Normative data on the diurnal pattern of the Karolinska Sleepiness Scale ratings and its relation to age, sex, work, stress, sleep quality and sickness absence/illness in a large sample of daytime workers. *Journal of sleep research*, 26(5), 559-566.

Supplementary Figure 12. Vigor Affect and Sleep quality questionnaires. (A) Results for each dimension of the vigor affect questionnaire. One participant has missing data in the morning due to a measurement error. (B) Results for the Karolinska Sleepiness Scale (KSS), a more standard approach to assess sleepiness as compared to the sleepy dimension of the vigor affect scale. We found a main time of day effect indicating a decreased KSS score from Evening (Eve) to Morning (Mor) (-2.71 ± 0.44 ; $F(1, 17) = 38.11$, $p < 0.001$). Gray lines are the mean response of each participant and black lines are the mean response across participants. (C) Left: Results for rating of sleep quality questions (mean \pm standard deviation). Right: Table which questions each acronym along the x-axis in the right figure corresponds to in German (white rows) and English (gray rows).

3. I found the description of the eye tracking system lacked several key aspects, including for instance calibration/validation procedures, sampling rate, accuracy etc.

Could the authors compare their deep-lab cut-based pupil assessment to more established eye trackers to indicate the validity?

We thank the reviewer for this important point. State of the art eye tracking systems have calibration and validation steps to be able to track gaze on a field of view, such as a monitor, and to obtain a 3d model of the eye for their pupil size and gaze estimates. We used DeepLabCut to track anatomical features of the eye and extracted pupil size from markers tracking 24 points around the pupil and iris circumference. For the purpose of this study, gaze data was not of interest, and we therefore did not conduct any classical calibration and validation steps. We did however visually inspect the labelling of the DeepLabCut markers on the targeted anatomical features. Further artifact removal was conducted using the guidelines and standardized open-source pipeline published by Kret and Sjak-Shie. Regarding sampling rate, video data from the eye tracker goggles had a varying sampling rate higher than 50Hz which was resampled to a fixed sampling rate of 50Hz. This is now stated on page 15 line 642: *Video data from the eye tracker goggles had a varying sampling rate higher than 50Hz. To calculate pupil size, the eye tracker video was first resampled to 50Hz, and processed using Python 3.8 and DeepLabCut 2.2*^{31,32}

As suggested by the reviewer, we compared our pupil assessment to an established eye tracker. The following test has been added to the Supplementary Figure 13. One participant sat alone in a room in a comfortable chair with their chin placed in a chin rest to ensure a stable head position. Their eyes were ~65 cm away from the eye tracker (Tobii TX300, Tobii Technology) that was positioned below the screen (240B7QPJ, resolution: 1,680 × 1,050 pixels; Philips) to allow for optimal eye tracking and measurement of pupil size. The participant was instructed to look at the fixation dot displayed in the center of the screen while the background display changed to one of three light intensities (darker, neutral, brighter) in a counterbalanced and randomized order, thereby inducing pupillary light reflexes at varying intensities. Pupil diameter for the right eye was recorded using the Tobii TX300 SDK for MATLAB v.3 and MATLAB 2013a). At the same time, the right eye was tracked using the same Pupil Core goggles used in the main study and Pupil Capture v3.4.0 software (Pupil Labs GmbH, Berlin, Germany). Using the video from Pupil Capture, pupil size for the right eye was calculated exactly as described on page 15 line 631 in the methods section. Pupil size derived from both eye trackers were downsampled to 25Hz, synchronized, normalized using z-scoring, and overlaid as shown in Supplementary Figure 6A. Pupil size derived using both methods were very strongly correlated ($R_{pearson}=0.995$, $p<0.001$, Supplementary Fig. 13B). The main limitation of this comparison is the varying intensity of infrared light emitted by the Tobii TX300. During the measurement, this varying intensity caused overexposure in the Pupil Core infrared cameras. As a result, the video footage became bleached, preventing the DeepLabCut algorithm from accurately tracking the anatomical features needed to measure pupil size. This is reflected in the missing orange values in Supplementary Figure 13A.

This new supplementary figure is referenced in page 15 line 662 as follows: *Further, temporally isolated samples with a maximum width of 50ms that border a gap larger than 40ms were removed (see Supplementary Fig. 13 for a comparison of our pupil assessment to an established eye tracker).*

Kret ME, Sjak-Shie EE. Preprocessing pupil size data: Guidelines and code. Behav Res Methods. 2019;51(3):1336-1342. doi:10.3758/s13428-018-1075-y

Supplementary Figure 13. DeepLabCut-based pupil size tracking versus established eye tracking. (A) Pupil size z-scored independently for both the established eye tracker assessment (blue) and the overlaid DeepLabCut (DLC)-based assessment (orange). (B) Pearson correlation of both eye tracking assessments. The black line indicates identical pupil size measurements with both assessments.

4. Some work in animals suggests that raw pupil size and its first derivative are differentially associated with cholinergic and noradrenergic neuromodulation [2]. The authors could complement their analyses by also including the first derivative of the pupil time series.

We thank the reviewer for this proposal. We have added a figure in the supplements (Supplementary Figure 7) showing the first derivative of pupil size during the microevents that we report in Figure 3. We limited the interpretation of the pupil size derivative to noradrenergic activity and did not include any possible cholinergic influence in the pupil derivative as Reimer et al. showed that moment-to-moment fluctuations (derivative measures) were tightly linked to NA, whereas it was the changes on a longer time-scale (non-derivative measures) that were closely related to the cholinergic system. We believe that this new figure will help convey our description of pupil dilations and constrictions that we discuss regarding Figure 3 on page 7 line 264: *We also calculated the derivative of pupil size during these events as it has been found to be tightly linked to the noradrenergic system and to cortical activity (Supplementary Fig. 7)*^{8,48}

Pupil size dynamics of each microevent depicted in Figure 3A-C are now further highlighted by this supplementary figure and we have therefore expanded the results section as follows.

For K-complexes on page 7 line 269: *This pupil size increase at onset was further highlighted by a significant positive change in the derivative of pupil size (Supplementary Fig. 7A).*

For sleep arousals on page 7 line 281: *This pupil dilation was descriptively larger than the dilations observed during K-complexes but also more variable across participants as the derivative of pupil size did not significantly deviate from zero (Supplementary Fig. 7B).*

For spindle trains on page 7 line 291: *Prior to the onset of the spindle train, there was a significant $11.51 \pm 1.46\%$ decrease in pupil size compared to baseline ($p < 0.05$) which corresponded to a significantly negative derivative of pupil size during this time period (Supplementary Fig. 7C). Pupil size remained relatively small throughout the spindle train and only after did it return to values observed 20s prior to the onset of the spindle train ($p < 0.05$), as reflected by the significantly positive derivative of pupil size after the spindle trains (Supplementary Fig. 7C).*

And on page 11 line 453: *Pupil size also constricted before the occurrence of spindle clusters and dilated after, as indicated by the significant pupil size derivative (Supplementary Fig. 7C). This was perhaps as a result of the reduced arousal levels mediated through reduced noradrenergic activity suggested to be required for spindles to occur*⁴⁴.

Supplementary Figure 7. Pupil size derivative during sleep events. (A-C) Derivative of pupil size that was normalized to the 5s prior the detected event in Figure 3A-C of the main manuscript: K-complexes, sleep arousals, and spindle trains (respectively). (C) Green vertical shading reflects the mean \pm 1 SEM of the median spindle train end across participants. Blue centre lines represent the group mean and shading around the centre lines the SEM. Gray lines are the mean response of each participant. Black horizontal lines mark significant differences from zero ($p < 0.05$). p -values are based on post-hoc t-test and are adjusted for multiple comparisons using Benjamini-Hochberg correction.

These novel findings are now integrated in the discussion on page 11 line 467 as follows:

In rodents, Eschenko et al. reported increased LC firing at the Down-to-Up state transition of slow waves, potentially leading to the synchronized depolarization of a large number of cells, which then influences the cortical networks generating this state transition⁵⁴. In addition, Osorio-Forero et al. show that relatively small LC activity surges lead to a phasic increase in low-frequency power, whereas larger ones cause a microarousal. This is evidence in favor of a functional analog of human K-complexes in rodents that reflects a graded arousal level in LC activity⁵². Here, we specifically focused on K-complexes and found rapid pupil dilations at their onset, pupil size dynamics that have been associated to phasic activity of noradrenergic axons⁸. Along these lines, Osorio-Forero et al. (2024) reported that transient LC activity increased delta power, possibly reflecting K-complexes that remain undefined in sleeping rodents⁵². Our results indicate that K-complexes involve a significant pupil dilation possibly reflecting transient LC activity time locked to these events. Interestingly, sleep arousals were also accompanied by pupil dilation and have previously been reported to be accompanied with bursts in LC activity¹⁸. However, this dilation was not as robust as that reported at the onset of K-complexes. This discrepancy may be due the unclear temporal alignment between pupil size dynamics and the detection of sleep arousal onsets as it is based on thresholding of changes in EEG power bands whereas with K-complexes it is based more precisely on the event related potential of the EEG signal.

5. Figure 5 reports evoked EEG responses in relation to pre-stimulation pupil size. Here the authors select 2 out of 4 bins with low and high pupil size. However, a lot of research in rodents[e.g., 3,4] shows (inverted) u-shaped associations with pupil size, which will not be detected with the current approach. I would therefore suggest using all the data collected, possibly even with a finer binning. More generally, I think the manuscript could be improved by extending their discussion of the relation between pupil-indexed neuromodulation and brain states beyond the sleep field.

We thank the reviewer for this truly thought-provoking comment. We agree with the reviewer, that it would be interesting to see whether the inverted u-shaped associations of evoked responses and pupil size previously shown in the awake state are also present in human sleep. We have now analyzed the two intermediary quartiles (bins) that were not included in the original analysis and report the findings in Supplementary Figure 11. Note that here quartile 1 and quartile 4 correspond to small and large quartiles in figure 5 in the main manuscript. We found an interaction effect indicating varying evoked responses in delta power from Stim_{pre} to Stim_{early} across pupil baseline conditions, $F_{\text{delta} \times \text{pupilrange}}(3, 49) = 4.15, p = 0.011$. Post-hoc analysis significant differences in evoked responses between quartile 1 and quartile 2 ($t(24.5) = 2.97, p = 0.040$), and between quartile 1 and quartile 4 ($t(24.5) = 2.87, p = 0.042$). No other contrasts were significant ($|t(24.5)| \leq 1.80, p \geq 0.306$). Regarding sigma power, we found no interaction effect between pupil baselines and sigma power at Stim_{pre} and Stim_{early}, $F_{\text{sigma} \times \text{pupilrange}}(3, 49) = 2.23, p = 0.097$, but did find a main effect of sigma power changes from Stim_{pre} to Stim_{early}, $F_{\text{sigma}}(1, 49) = 20.43, p < 0.001$. Every quartile except quartile 4, showed increased sigma power at Stim_{early} compared to Stim_{post}.

These findings are reported in the main text of the manuscript on page 9 line 352 as follows: *When we included all quartiles into which the N2 verum trials were separated, the aforementioned relationship of increasing evoked response in power with decreasing pupil size at baseline became nonlinear (Supplementary Fig. 11). Increased evoked response in sigma and delta power occurred in intermediary quartiles but not as robustly as in the small pupil baseline conditions.*

These findings are now integrated into the discussion and compared to the relation between pupil-indexed neuromodulation and brain states beyond the sleep field on page 12 line 486. This now reads: *According to previous research, we found auditory stimulation to increase delta and sigma power dynamics^{50,55-61}. These delta responses are believed to involve enhanced slow waves or K-complexes while sigma responses involve enhanced spindle activity^{62,63}. However, this increase in delta and sigma power was indeed not present when pupil size was high (largest quartile) prior to stimulation onset (Fig 5, Supplementary Fig. 11). Therefore, a minimum pupil size might be a prerequisite for the induction of sensory-evoked delta and sigma enhancement, with the optimal evoked response potentially occurring at low arousal levels. This is in contrast to inverted U-shaped relationships between pupil-indexed arousal levels and neuromodulation that have been observed during wakefulness in rodents^{64,65}, and therefore adds additional complexity to our current understanding of neuromodulation across the entire arousal level continuum.*

These additional findings reveal that auditory stimulation enhances delta and sigma

The limitations of our interpretation have been emphasized by suggesting future research directions on page 12 line 514: *While finer binning of pupil size before stimulation windows could have provided further insights into the relationship between sensory-evoked responses and specific arousal levels, the auditory stimulation analysis was constrained by the number of trials and subjects with valid pupil data during stimulation windows. Monitoring pupil size online and applying sensory stimulation based on ongoing pupil size dynamics in a larger sample could provide a more representative sample of evoked response across all arousal levels of NREM. Future studies could use such an approach to identify optimal arousal levels to maximize target responses while mitigating the likelihood of awakenings.*

Supplementary Figure 11. EEG responses to auditory stimulation at pupil size baselines separate into quartiles. (A) Sigma power 5s before (Stim_{pre}) and during the first 5s (Stim_{early}) of verum stimulation trials sorted from trials with smallest pupil size at Stim_{pre} (quartile 1) to trials with largest pupil size at Stim_{pre} (quartile 4). Note that here quartile 1 and quartile 4 correspond to small and large quartiles in figure 5A in the main manuscript. We found no interaction effect between pupil baselines and sigma power at Stim_{pre} and Stim_{early}, $F_{\text{sigma} \times \text{pupilrange}}(3, 49) = 2.23$, $p = 0.097$, but did find a main effect of sigma power changes from Stim_{pre} to Stim_{early}, $F_{\text{sigma}}(1, 49) = 20.43$, $p < 0.001$. Every quartile except quartile 4, showed increased sigma power at Stim_{early} compared to Stim_{pre}. (B) Delta power 5s before (Stim_{pre}) and during the first 5s (Stim_{early}) of verum stimulation trials sorted from trials with smallest pupil size at Stim_{pre} (quartile 1) to trials with largest pupil size at Stim_{pre} (quartile 4). Note that here quartile 1 and quartile 4 correspond to small and large quartiles in figure 5B in the main manuscript. We found an interaction effect indicating varying evoked responses in delta power from Stim_{pre} to Stim_{early} in across pupil baseline conditions, $F_{\text{delta} \times \text{pupilrange}}(3, 49) = 4.15$, $p = 0.011$. For posthoc analysis we quantified the evoked response as the difference between Stim_{early} and Stim_{pre} and derived post-hoc p-values on the differences in evoked response between different pupil baseline conditions using Satterthwaite's method from the R package lmerTest and corrected for multiple comparisons with the Hochberg method using the R package emmeans. Post-hoc analysis showed significant differences in evoked responses between quartile 1 and quartile 2 ($t(24.5) = 2.97$, $p = 0.040$), and between quartile 1 and quartile 4 ($t(24.5) = 2.87$, $p = 0.042$). No other contrasts were significant ($|t(24.5)| \leq 1.80$, $p \geq 0.306$). The number next to the gray lines correspond to the participant number. The stimulation protocol V0 was used in 2 and 4, and V1 for the rest. The horizontal black lines connecting pupil size ranges are the interaction effects between pupil size range and Stim_{pre}-Stim_{early} power values. p-values are based on post-hoc t-test and are adjusted for multiple comparisons using Benjamini-Hochberg correction.

Related to this analysis, it would be interesting to also include associations between baseline and evoked pupil dilation (cf. e.g., [5]).

Furthermore, we investigated the associations between baseline and evoked pupil dilation and found main effects of pupil size at baseline ($F(1, 21)=54.94, p<0.001$) and evoked pupil dilation ($F(1, 21)=9.60, p=0.005$) but no interaction effect ($F_{\text{baseline} \times \text{dilation}}(1, 21)= 0.042, p=0.829$), suggesting that the evoked pupil dilation was not significantly different at different baselines. These findings are reported in Supplementary Figure 10 and interpreted in the main text on page 9 line 344: *We then investigated the associations between baseline and evoked pupil dilation and found main effects of pupil size at baseline ($F(1, 21)=54.94, p<0.001$) and evoked pupil dilation ($F(1, 21)=9.60, p=0.005$). We found no interaction effect between baseline and evoked pupil dilation, suggesting that the evoked pupil dilation was not significantly different at different baselines ($F_{\text{baseline} \times \text{dilation}}(1, 21)= 0.042, p=0.829$; see Supplementary Fig. 10).*

Supplementary Figure 10. Evoked pupil response for small and large pupil size at baseline. Pupil size 5s before ($Stim_{pre}$) and during the first 5s ($Stim_{early}$) of verum stimulation of trials with large (highest quartile) and small pupil baselines (lowest quartile) to group into high and low arousal level trials. We found main effects of pupil size at baseline ($F(1, 21)=54.94, p<0.001$) and evoked pupil dilation ($F(1, 21)=9.60, p=0.005$) but no interaction effect $F_{\text{baseline} \times \text{dilation}}(1, 21)=0.042, p=0.829$.

6. In their discussion, the authors write that locus coeruleus neuromodulation may be involved in several of the reported sleep metrics. (e.g., line~364, 396) I would like to encourage the authors to be more specific and state the mechanisms of **how** neuromodulation shapes sleep rhythms.

We thank the reviewer for raising this important question to our attention. With regards to spindles, it has been reported by Osorio-Forero et al. that “NA induces a slowly decaying membrane depolarization through activation of both α_1 or β receptors in thalamocortical and thalamic reticular neurons, which retards the re-engagement of these cells in sleep-spindle generation.” We have extended our discussion on page 11 line 428 to include a summary of this mechanistic explanation of the LC in spindle dynamics: *Furthermore, it has been shown that also LC-NA activity exhibits pronounced ISFs during NREM sleep and that high LC-NA activity is associated with a decreased occurrence of sleep spindles. This may be due to the depolarization of thalamocortical and thalamic reticular neurons, which delays their re-engagement in sleep spindle generation*³

While it is challenging to specify the neuromodulatory mechanisms of the locus coeruleus (LC) and K-complexes in humans, we can infer their characteristics through shared neurophysiological signals between humans and rodents. In rodents, the neuromodulatory mechanisms of sleep are better understood. However, a noteworthy limitation of this approach is the lack of consensus on whether rodents exhibit K-complexes. Despite this, the strong relationship between K-complexes and slow waves allows us to infer that the LC-NA neuromodulatory dynamics during slow waves in rodents may be similar to those during human K-complexes. With this in mind, we have extended our discussion on page 11 line 467 to include the interpretation from Eschenko et al.: *In rodents, Eschenko et al. reported increased LC firing at the Down-to-Up state transition of slow waves, potentially leading to the synchronized depolarization of a large number of cells, which then influences the cortical networks generating this state transition*⁵⁴. *In addition, Osorio-Forero et al. show that relatively small LC activity surges lead to a phasic increase in low-frequency power, whereas larger ones cause a microarousal. This is evidence in favor of a functional analog of human K-complexes in rodents that reflects a graded arousal level in LC activity*⁵².

Osorio-Forero A, Cherrad N, Banterle L, Fernandez LMJ, Lüthi A. When the Locus Coeruleus Speaks Up in Sleep: Recent Insights, Emerging Perspectives. *Int J Mol Sci.* 2022;23(9). doi:10.3390/ijms23095028

Eschenko O, Magri C, Panzeri S, Sara SJ. Noradrenergic neurons of the locus coeruleus are phase locked to cortical up-down states during sleep. *Cereb Cortex.* 2012;22(2):426-435. doi:10.1093/cercor/bhr121

Minor comments

1. Introduction: The accessibility of the manuscript could be increased by including brief introductions for the different sleep metrics (e.g., spindles, ~ line 66). For instance, k-complexes are only introduced at the end of the paper (~ line 396) – providing this information earlier would help readers who are not well versed in the sleep field.

We completely agree with the reviewer. To make the manuscript more accessible, we've added the definition of K-complexes earlier on page 5 line 206: [...] *K-complexes, NREM microevents representing highly synchronized bottom-up slow waves that potentially involve LC activation and the promotion of highly synchronized cortical activity^{42,43} and have been argued to reflect arousal level fluctuations¹.*

sleep spindles on page 2 line 79: *In mice, ISFs in NA-driven arousal level are linked to the thalamocortico-cortical activity in the forebrain. This activity includes the emergence of sleep spindles (transient bursts of sigma oscillations in the 12-16 Hz range) that are regulated by NA-dependent control of the membrane potential of thalamic neurons^{3,17,18}. NA levels show an anti-correlation with spindle activity in the cortex and a positive correlation with heart rate (HR).*

and sleep arousals on page 7 line 253: *According to these criteria, sleep arousals are characterized by abrupt changes in EEG frequency, suggestive of an awake state, which must be accompanied by increases in EMG amplitude during REM.*

2. Results: Since there was considerable eye tracking data loss, the authors should report (can be SI) if data loss was uniformly distributed across the pupil/sleep stages.

We thank the reviewer for this valuable input. We have now added the proposed information in the form of Supplementary Figure 1. The results suggest that data loss is uniformly distributed across sleep stages but is most prominent during wake. This is presumably explained by the fact that there are more eye movements and likely more occurrences of Bell's phenomenon, the upward rolling of the eyes when closing the eyelids. The figure is referenced on page 4 line 142 in the main manuscript.

Note that Supplementary Figure 1 in the original submission is now Supplementary Figure 4 as it is referenced in the main manuscript later on.

Supplementary Figure 1. Pupil size quality. Percentage of time during eye tracking with valid pupil size measurements for each sleep stage. W: wakefulness (yellow); N1: NREM stage 1 (gray); N2: NREM stage 2 (green); N3: NREM stage 3 (blue); R: rapid eye movement (REM, orange).

3. Results: “Arousal levels can be used to define sleep depth and have been shown to depend on the sleep macro-architecture” – seems like there is a citation missing here.

We thank the reviewer for pointing this error out. We have now added the following citations:
 - Lendner, J. D., Helfrich, R. F., Mander, B. A., Romundstad, L., Lin, J. J., Walker, M. P., ... & Knight, R. T. (2020). An electrophysiological marker of arousal level in humans. *elife*, 9, e55092.

- Höhn, C., Hahn, M. A., Lendner, J. D., & Hoedlmoser, K. (2024). Spectral Slope and Lempel–Ziv Complexity as Robust Markers of Brain States during Sleep and Wakefulness. *Eneuro*, 11(3).

4. Figure 1 B could be improved by including and labeling the different parameters that were used to estimating pupil size. **Added.**

5. Results/Sl: I could not find the statistical results (p-value) related to the cross-correlations, could these be added, along with marking the significant lags in the figure (incl. a control for multiple comparisons)?

We thank the reviewer for raising this point. To our knowledge, it is uncommon to report p-values of cross-correlations in sleep research (e.g. Osorio-Forero et al. 2021 and Kjaerby, Andersen et al. 2022) and we have not encountered an established method for doing so. However, we have attempted to calculate the significance of the cross-correlations between the mean sigma and pupil size as well as mean sigma and heart rate during infraslow cycles across participants (black, blue, and red lines in Figure 2D,E). We used the package *testcorr* in R, which implements robust procedures for testing the significance of the cross-correlation in bivariate data. The null hypothesis is that there is no correlation between the two signals being compared. We have

added Supplementary Figure 5 to show that indeed the cross-correlation values we report are significant ($p < 0.001$) and reference this supplementary figure in the main text on page 6 line 221 and line 227.

Osorio-Forero A, Cardis R, Vantomme G, et al. Noradrenergic circuit control of non-REM sleep substates. *Curr Biol.* 2021;31(22):5009-5023.e7. doi:10.1016/j.cub.2021.09.041

Kjaerby C, Andersen M, Hauglund N, et al. Memory-enhancing properties of sleep depend on the oscillatory amplitude of norepinephrine. *Nat Neurosci.* 2022;25(August). doi:10.1038/s41593-022-01102-9

Dalla, V., Giraitis, L., & Phillips, P. C. B. (2021). testcorr: Testing Zero Correlation (Version 0.2.0) [R package]. Retrieved from <https://CRAN.R-project.org/package=testcorr>

Supplementary Figure 5. Significance of cross-correlations during N2. (A) Cross-correlation of mean sigma power (source) and mean pupil size across participants (black line) with the lag being with respect to the cycle percentage of the ISF. (B) Cross-correlation of mean sigma power (source) and mean heart rate across participants (black line) with the lag being with respect to the cycle percentage of the ISF. We used the package *testcorr* in R, which implements robust procedures for testing the significance of the cross-correlation in bivariate data. The null hypothesis is that there is no correlation between the two signals being compared. The robust p -values below 0.001, indicating significant correlations are reported in the form of black horizontal lines at the bottom of each figure.

6. Results: “Arousals from sleep as defined by AASM guidelines are brief perturbations to sleep which are characterized by large and sudden changes in the theta, alpha, or beta frequencies in the EEG power spectra, which can be accompanied by EMG activation.” – this section could be clearer when including the directionality of spectral changes.

We agree with the reviewer in that the section could be made clearer by including the directionality of spectral changes. Therefore, we have amended the quoted section on page 7 line 276 to the following: “Arousals (Fig. 3B) from sleep as defined by AASM guidelines are brief perturbations to sleep which are characterized by large and sudden increases in the theta, alpha, or beta frequencies in the EEG power spectra, which can be accompanied by increased EMG activation.”

7. Figure 3 E: While I do not have much experience with heart rate data, I was surprised by the variance difference between Fig 3 E relative to 3D/F. Could the authors elaborate on this?

We thank the reviewer for pointing out this discrepancy in variance. These variance differences are likely driven by the autonomic differences across the vigilance states between sleep and the bout of wakefulness that is an arousal. Furthermore, Sforza et al. report heart rate changes to occur before the onset of arousals. Therefore, the unclear temporal alignment between cardiovascular responses and the detection of arousal onsets, based on thresholding of changes in EEG power bands, may further explain the variance difference between Figure 3 E relative to Figure 3 D,F.

Sforza, E., Jouny, C., & Ibanez, V. (2000). Cardiac activation during arousal in humans: further evidence for hierarchy in the arousal response. *Clinical Neurophysiology*, 111(9), 1611-1619.

8. Line 618, repeated word: arousal. Corrected.

Literature

1 Berridge, C.W. (2008) Noradrenergic modulation of arousal. *Brain Res. Rev.* 58, 1–17

2 Reimer, J. et al. (2016) Pupil fluctuations track rapid changes in adrenergic and cholinergic activity in cortex. *Nat. Commun.* 7, 13289

3 McGinley, M.J. et al. (2015) Cortical membrane potential signature of optimal states for sensory signal detection. *Neuron* 87, 179–192

4 McGinley, M.J. et al. Waking State: Rapid Variations Modulate Neural and Behavioral Responses. , *Neuron*, 87. 23-Sep-(2015) , Cell Press, 1143–1161

5 Collins, L. et al. (2021) Vagus nerve stimulation induces widespread cortical and behavioral activation. *Curr. Biol.* 31, 2088–2098.e3

.....

Reviewer #3 (Remarks to the Author):

Thank you for inviting me to review this manuscript, which used an inventive approach – namely, taping a participant’s eyelid open and then having them fall asleep – to track pupillary dynamics across sleep stages. The authors then compared these recordings to the power within different frequency bands estimated from scalp electrophysiology and also in response to auditory tones. These data provide evidence that the pupil can be used to track responses to stimuli during sleep.

We are grateful for the reviewer’s constructive feedback, which motivated us to undertake further analyses and refine our manuscript.

* The authors may wish to compare their results to other studies that have tracked pupil diameter across sleep stages in animals, such as Tsunematsu et al., 2020 eLife.

We thank the reviewer for raising this important point. Our results do indeed share parallels with animal studies that measured pupil size during sleep, including the work referenced by the reviewer. We have added this reference alongside the animal work that was already cited in the original submission in the results section. Furthermore, we have made a figure similar in format to those reported by Yüzgeç et al. 2018, Kobayashi et. al 2023, and Tsunematsu et al., 2020, in Supplementary Figure 2. One can find these edits in the results section on page 4 line 157: *For a more detailed quantification of the distribution of pupil size across different sleep stages, see Supplementary Fig. 2A. Lower pupil size during NREM and REM sleep when compared to wake in addition to the variable distribution of the size of the pupil within sleep stages is consistent with previous pupillometry findings in sleeping rodents, where NREM sleep is not subdivided into stages^{19,36,37}*

Supplementary Figure 2. Probability density plot of pupil size. (A) Pupil size probability for each sleep stage. (B) Pupil size probability for each sleep stage but with NREM sleep stages pooled together (N1+N2+N3). W: wakefulness (yellow); N1: NREM stage 1 (gray); N2: NREM stage 2 (green); N3: NREM stage 3 (blue); R: rapid eye movement (REM, orange), N1+N2+N3: NREM stages 1-3 pooled together (dark gray).

We have also changed the discussion on page 10 line 402 to reflect the new findings: *Note that there is no currently established subdivision of rodent NREM sleep into stages. The small pupil size that characterizes both REM sleep and the N3 stage of NREM sleep has remained undetected*

until now. For a more direct comparison with pupil size changes across sleep states in rodent models, we pooled NREM sleep stages together (Supplementary Fig. 2B). This revealed a coarser distribution in pupil size during NREM which largely overlaps with pupil size values during REM and minutely with wakefulness. It is remarkable that, similar to pupil diameter distributions, LC activity ranges in rodent NREM sleep also overlap with the ones of REM sleep and of wakefulness⁵². The positive skewness of the pooled data, as a result of N3 samples, is also visible in rodent sleep^{19,36,37}. It is therefore tempting to speculate that the low pupil size values in NREM rodent sleep is a distinct brain state with analogous properties to human N3 sleep.

* Sorry if I missed this, but how were the differences in ambient light levels across sleep and wake controlled for?

All reported eye tracking measures were recorded in a dark room where the eyes were only exposed to the infrared light from the eye tracker. Therefore, all vigilance states had exactly the same lighting conditions. This has now been clarified on page 3 line 127: *Data acquisition started once the room was completely dark, except for the infrared light source emitted by the pupillometer, ensuring all vigilance states had exactly the same lighting conditions.*

* Figure 1H – was there a reason that the authors didn't standardize the HR patterns across subjects (as they did for pupil size and spectral slope)?

We thank the reviewer for this comment. The reviewer might have been misled by the arbitrary units shown in Fig1 F,G. Pupil size and slope weren't standardized. As mentioned in the methods, pupil size was normalized to each individual's iris size. The slope value is simply the slope of the power spectrum in the log-log scale.

* Figure 1 – rather than plotting a single value for each brain state per subject, I wonder whether it would be more informative to plot the distribution of pupil sizes as a function of brain state. This will help to provide a deeper intuition for the range of pupillary dilation across different brain states. Given that the pupillary traces shown in 1C are distinctly non-linear, it is difficult to know whether the mean value within each state is an effective summary of the dependent variable. The fact that this measure is used to make inferences later in the manuscript makes this an important baseline issue to resolve in order to ensure robust interpretability of the authors work.

We thank the reviewer for raising this point. We do agree that the proposed plot provides a deeper intuition for the range of pupillary dilation across different sleep stages. We believe that our current figure holds additional information as it shows a consistent drop in pupil size across sleep stages at the participant-level. This information, which may be more clinically relevant, would be lost if we replaced it with the proposed plot. We have therefore made the proposed plot in the form of Supplementary Figure 2 and reference it in the main text on page 4 line 159 and page 10 line 406. Considering the distribution of pupil size for each sleep stage, we believe that this figure supports the use of the mean value within each sleep stage as an effective summary of the dependent variable.

* Figure 2 – did the authors account for differences in the spectral content of the signals they were comparing? By eye, the pupils + HR signals are quite low-frequency relative to the sigma power fluctuations, whereas the K-complexes are relatively high-frequency.

The differences in spectral content between lower-frequency pupil+HR signals and the higher-sigma power fluctuations was, in part, accounted for in the low-pass filtering of sigma power as described in the methods section.

A higher frequency signal from the frontal electrode was needed to detect K-complexes as recommended by the AASM guidelines and illustrated in Figure 2C. However, this signal was only used for the detection of the K-complex. All statistical analysis were based on the likelihood of detecting a K-complex during an infraslow fluctuation of sigma power activity, a metric that is marginally influenced by the spectral content of the Fpz signal.

Critically, our statistical analyses for Figure 2 G,H,I used the mean value of each signal for each participant for discrete portions of the infraslow cycle (Trough, Rise, Peak, Fall). Therefore the differences in the spectral content between signals was accounted for by comparing them with regards to their average for a given portion of the infraslow cycle.

* Figure 3 - The variance in the plots is quite large (as shown by the gray lines, which is excellent), though the use of S.E. rather than S.D. to define the error bars might appear somewhat misleading to the reader.

We thank the reviewer for this comment. We believe the gray lines is the most informative visualization of variability across participants as it depicts from which participants the group variability originates from. If we used S.D., then the gray lines would be occluded further by the S.D. shading, so we decided to use S.E. as it is also a common approach for depicting the variance of a signal.

* L286 – The correct name of the correction used is “Benjamini”, not “Benjamin” **Corrected**

* Did the auditory stimulations during sleep rouse the participants?

We thank the reviewer for bringing up this important point. We now added Supplementary Table 8 including the number of arousals, as defined by the AASM guidelines, during auditory stimulation (Verum) and during Sham. The last two columns show the percentage of stimulation windows for both Verum and Sham where an arousal was detected. These results show that the auditory stimulations like sham periods co-occurred with few to no arousals. Auditory stimulation during sleep does indeed elevate arousal levels as reflected by a significant increase in pupil size with tones played when compared to sham (Figure 4A,D). We now mention these findings in the main text on page 16 line 698: *See Supplementary Table 8 for an overview of what participants were included in each stimulation protocol and the amount of arousals that co-occurred with the ON windows. Auditory stimulation does indeed elevate arousal levels, as reflected by a significant increase in pupil size with tones played when compared to sham (Figure 4A,D). We therefore excluded ON windows where tones were played from Figures 1, 2, and 3 and their corresponding statistical analysis.*

Participant number	Stimulation protocol	Stimulation analysis included in	Number of sham windows	Number of verum windows	Number of arousals during sham windows	Number of arousals during verum windows	Sham windows with arousals (%)	Verum windows with arousals (%)
1	V0	None	-	0	-	0	-	0
2	V0	Figure 5 All	-	303	-	4	-	0.33
3	V0	None	-	85	-	3	-	1.17
4	V0	Figure 5 All	-	94	-	1	-	1.06
5	V0.5	None	12	14	0	0	0	0
6	V0.5	None	0	0	0	0	NaN	NaN
7	V0.5	None	1	1	0	0	0	0
8	V1	Figure 4 All and Figure 5 All	74	72	1	1	1.35	1.39
9	V1	Figure 4 All	15	14	0	0	0	0
10	V1	Figure 4 All and Figure 5 All	66	64	0	0	0	0
11	V1	Figure 4 All and Figure 5 All	46	45	0	0	0	0
12	V1	Figure 4 All	56	55	0	1	0	1.82
13	V1	Figure 4 All and Figure 5 All	48	44	0	0	0	0
14	V1	Figure 4 All and Figure 5 All	47	45	0	0	0	0
15	V1	Figure 4 All	55	54	0	0	0	0
16	V1	Figure 4 B,C,E,F	15	15	1	0	6.67	0
17	V1	Figure 4 B,C,E,F	8	7	0	0	0	0

Supplementary Table 8. Stimulation protocols. Description of what participants were included in each stimulation protocol and corresponding analysis in Figure 4 and Figure 5. The number of sham and verum windows and the number of arousals co-occurring with these windows are reported.

* Discussion: The statement “Therefore, we propose that pupil size dynamics might represent a non-invasive way to track arousal levels” disregards a large literature that has already made similar inferences. I suggest re-wording the main conclusion to be more specific to the results of this experiment.

We agree with the reviewer and have changed our interpretation accordingly. Our statement has been modified to be more specific to human sleep and to refer to rodent literature with similar inferences. The statement now reads as follows on page 10 line 395: *Pupil size dynamics during human sleep not only correlate with cortical markers of arousal level dynamics, but are also in line with arousal level dynamics reported in rodent sleep. Therefore, pupil size dynamics might represent a non-invasive way to track arousal levels in human sleep*

We would like to thank the editor and the reviewers for once again taking the time to assess our work. We modified parts of the manuscript based on the reviewers' feedback. Below, we provide a point-by-point response to each comment (in blue).

Reviewer #1 (Remarks to the Author):

1. Auditory Stimulation Protocol (Methods):

(Page 2, line 680 in the manuscript)

Issue: The reviewer found the auditory stimulation protocol confusing, specifically about the 10s on and 20s off cycle, and whether this was EEG event-based.

Feedback: The response clarifies the stimulation protocol well. Although citations for the EEG feedback developed you are given, could you please add a sentence or two to make the detection processes could more clear.

We thank the reviewer for suggesting a clearer description of the detection processes. We believe that expanding the following sentences (in bold) on page 16 line 699 further clarify the protocol: *In V0 of the stimulation protocol, the onset of the ON window was conditional on the detection of the ascending phase of a slow wave during NREM sleep, **using a first-order phase-locked loop on the 0.1-40 Hz bandpass filtered signal from electrode Fpz.***^{55,88} *In V0.5 and V1, the onset of the ON window was conditional on the detection of an increase or a decrease in sigma power during NREM sleep inspired by Cardis et al.*⁴⁴ **This consisted of estimating sigma power using the square of the 12-15Hz bandpass filtered signal from electrode Cz and comparing current sigma power with the previous 28 s.**

Citation 88 is the original paper describing the PLL protocol to detect the phase of the slow wave:

Ferster, M. L., Da Poian, G., Menachery, K., Schreiner, S. J., Lustenberger, C., Maric, A., Huber, R., Baumann, C. R., & Karlen, W. (2022). Benchmarking Real-Time Algorithms for In-Phase Auditory Stimulation of Low Amplitude Slow Waves With Wearable EEG Devices During Sleep. *IEEE transactions on bio-medical engineering*, 69(9), 2916–2925.

<https://doi.org/10.1109/TBME.2022.3157468>

2. Participant Count and Protocol Variation:

(Supplementary Table 8 on page 8 of the main text)

Issue: Reviewer wanted clarification on the different stimulation approaches (verum vs. sham) and their effects on Figures 1–3.

Feedback: This concern has been well addressed by providing details on participant numbers in each protocol and generating a new table in the supplement. However, the question regarding the exclusion of stimulation affecting arousal level is not addressed enough. This should be discussed as a potential limitation if it cannot be a priori excluded. Also, since the usage of the terms “sleep arousals” (as determined by PSG) and “arousal levels” does not seem to be stringent throughout the manuscript, the differential (or similar) effect of stimulation on both measures should be more clearly made.

Although we excluded the stimulation phase and the period during which tones were emitted was relatively short compared to the total duration of NREM sleep, it is still possible that stimulation had after-effects on arousal levels, which cannot be ruled out a priori. We have therefore added this as a limitation on page 14 line 608 that includes a separate statement for sleep arousals: *This*

exclusion minimized the sensory evoked changes in arousal levels in the analyses. Although the tones were played for a short time compared to the total NREM sleep, and sleep arousals during the stimulation were rare and similar between sham and verum periods, the possibility that auditory stimulation affected arousal levels beyond the stimulation periods cannot be ruled out.

3. Arousal Definition:

(Page 2, line 53 (arousal levels) and page 7, line 253 (sleep arousals))

Issue: Reviewer requested a clear definition of “arousal” and how different types of arousal are related.

Feedback: This has been addressed well, however comprehension on how “arousal level” is defined would benefit greatly from a more concise description. For instance, it seems on the one hand pupil size is used as a proxy for arousal level (p. 8, l. 317), on the other hand, it is assumed that pupil size reflects arousal level, and the effect on EEG activity is the output measure (p. 8, l. 335).

We agree that a more concise definition of arousal level could help improve clarity. We have attempted to do so by (1) A more consistent use of our definition of arousal level and its relationship to pupil size has been applied across the manuscript by referring pupil size dynamics as an index of arousal level dynamics. (2) By consistently referring to sleep arousals derived from PSG data using the AASM guidelines throughout the manuscript. (3) Reformulating the first paragraph of the introduction that introduces the concept of arousal level and its relation to pupil on page 2 line 53 as follows:

Arousal level can be viewed as a continuum of activation of cortical and subcortical networks that influences how stimuli are processed across all vigilance states. During wake, arousal levels are closely tied to alertness, while during sleep, according to rodent studies drivers of arousal levels are strongly suppressed during non-rapid eye movement (NREM) and rapid eye movement (REM) sleep^{1,2}. Central drivers of arousal levels include brain areas such as the noradrenergic (NA) locus coeruleus (LC)³, the cholinergic nucleus basalis of Meynert⁴, serotonergic neurons in the dorsal raphe⁵, and orexin/hypocretin-producing neurons in the lateral hypothalamus⁶. Activity changes in these arousal level-regulating brain areas have further been linked to pupil size changes, meaning changes in pupil size can index their activity⁷⁻¹⁰. Other drivers of arousal levels that project to these nuclei include neuropeptide S¹¹, the cortex¹², and the autonomic nervous system¹³. Arousal level dynamics can be large and sustained, resulting in transitions from low vigilance states of sleep, such as NREM sleep, to larger states manifesting as wakefulness, or from low to even lower states manifesting as REM sleep. Alternatively, they can be subtle and temporary, detected primarily by neurophysiological markers such as K-complexes^{14,15}, spindles¹⁶, or sleep arousals, defined as very short bouts of wakefulness manifesting as abrupt but temporary changes in polysomnographic (PSG) signals¹⁷. However, little is known about whether similar arousal level dynamics exist also during human sleep and how they relate to sleep macrostructure and microstructural events. To answer this question, we measured pupil size to index arousal levels (Fig. 1), together with PSG and electrocardiography (ECG).

4. Auditory Stimulation and Arousal Levels:

(Page 14, line 601 in the manuscript)

Issue: There was confusion about how stimulation might affect arousal levels.

Feedback: The below definition was given (p. 3, 3rd to last paragraph), I suggest writing “...which in REM sleep must be accompanied by increases in EMG amplitude”

This was edited accordingly.

Reviewer #2 (Remarks to the Author):

I thank the authors for their thorough revision, which addressed my concerns. Below, I provide some minor suggestions that the authors may address:

Regarding my initial major comment #2 (open vs closed eye sleep differences): I agree that without a proper control night, the provided comparison is not conclusive (perhaps the authors even have access to such data?). To provide a reference for future follow up studies, I would nonetheless suggest including the figure the authors added to their rebuttal in the supplementary information.

We thank the reviewer for raising this point. Unfortunately we do not have a control night of these participants. However, we agree that including the figure added to the rebuttal as supplementary information would be beneficial for future follow up studies. We have now added the figure as part of Supplementary Figure 1 and reference it in the main manuscript on page 3 line 121 as follows: *All participants could tolerate having their eyes taped open and were able to fall asleep. While there were changes in sleep macrostructure between taping the right eye open in the first half of the night and having it closed in the second half of the night, these changes were consistent with a typical night's sleep, and no major disturbances were found from sleeping with the eye open (see Supplementary Fig. 1A for sleep macrostructural difference when the right eye is open versus closed). However, to truly answer this question, one would have to run a larger study with multiple sleep sessions in the laboratory.*

Supplementary Figure 1. Overnight sleep and pupil quality. (A) Percentage of time spent in each sleep stage in the sleep window when the right eye is taped open (Eye Open) versus in the sleep window after the tape is removed and the eye can be closed (Eye Closed). N1: NREM stage 1; N2: NREM stage 2; N3: NREM stage 3; R: rapid eye movement (REM). p-values are based on post-hoc t-test and are adjusted for multiple comparisons using Benjamini-Hochberg correction. (B) Percentage of time during eye tracking with valid pupil size measurements for each sleep stage. W: wakefulness (yellow); N1: NREM stage 1 (gray); N2: NREM stage 2 (green); N3: NREM stage 3 (blue); R: rapid eye movement (REM, orange). The participant showing the most data loss across sleep stages (lowest and flattest gray line) was only included for pupil-related analyses during wake (Figure 1F, Figure 1I, and Supplementary Figure 2).

- Regarding my initial minor comment #2 (data loss): One participant shows very little valid data (25 % max during W, dropping to values close to zero for the remaining stages), was this participant excluded from analyses? Would be good to clarify this in the legend.

Due to the data loss, the participant was not included in pupil-related analyses regarding NREM or REM. They were only included in pupil-related analyses regarding wake (Figure 1F, Figure 1I, and Supplementary Figure 2).

We have now added the following to the end of Supplementary Figure 1 legend: *The participant showing the most data loss across sleep stages (lowest and flattest gray line) was only included for pupil-related analyses during wake (Figure 1F, Figure 1I, and Supplementary Figure 2).*

- Regarding the analyses reported in fig 5, this paper showing state-dependent effects of locus coeruleus stimulation on cortical electrophysiology in anesthetized rodents could be useful: <https://pubmed.ncbi.nlm.nih.gov/29093170/>

We thank the reviewer for this insightful suggestion. The referenced paper does indeed support the interpretation of our findings and has therefore been referenced on page 12 line 497: [...] *and therefore adds additional complexity to our current understanding of neuromodulation across the entire arousal level continuum*⁶⁶.

- I enjoyed reading a recent related publication, demonstrating closed-eye pupil tracking using an alternative method that the authors could qualitatively compare to their method:

<https://www.ncbi.nlm.nih.gov/pmc/articles/PMC11303404/>

The referenced method offers a novel approach for estimating pupil size. While no sleep measurements are reported, it shows promise for use in sleep studies without taping the eyes open. The authors validated the technique by comparing indirect measurements from a closed eye to true measurements from the open eye during a pupillary light reflex test. We look forward to seeing this method applied in the sleep field to potentially measure the subtle pupil size dynamics we report in our manuscript. For that reason we now mention the referenced paper on page 13 line 536 as follows: *Recent advancements in estimating pupil size behind closed eyelids hold promise for the field of sleep research, which could build on our findings provided they can accurately capture subtle pupil size dynamics during sleep*⁶⁸.

Reviewer #3 (Remarks to the Author):

The authors have adequately addressed my concerns.

We would like to thank the editor and the reviewers for once again taking the time to assess our work. We modified parts of the manuscript based on the reviewers' feedback. Below, we provide a point-by-point response to each comment (in blue).

Reviewer #1 (Remarks to the Author):

The authors have addressed all my concerns nicely. I just have a few minor comments:

l.691-692: I believe there is a typo in the unit of this sentence:

“...level. This burst was presented every second for a duration of 10 seconds (ON window) followed by a period of silence (OFF window) where no auditory stimulation was presented....”

l.299 ‘N2 that were tones were’, omit ‘that’

l. 300, the term “general N2 levels” is first used here. Since this description ‘slips in’ as a frequently used description, but is not clearly defined, I suggest to add in brackets here e.g., (i.e., pupil size during N2, excluding the 5 s before and 5 s after an auditory stimulation).

Discussion

Although such analyses would go beyond the scope of this manuscript, in the discussion (around l.423 ff) it should be addressed that other parameters derived from the ECG, and thus reflecting autonomic nervous system activity may reveal a/another relationship to sleep stages/ISF and pupillometry (e.g., Naji et al. 2019, Neurobiology of Learning and Memory).

We thank the reviewer for once again reviewing our manuscript with great detail and for proposing to extend the discussion to elaborate on the potential of ECG-derived measures to further understand the relationship between autonomic nervous activity, sleep, and pupillometry. We believe that expanding the following sentences (in bold) on page 8 line 338 emphasis this point: This suggests that HR variations may be too moderate to capture arousal level fluctuations or that this measure seems to be less closely linked to arousal level dynamics indexed by spectral slope. **However, other ECG-derived measures not investigated here may further elaborate the role of autonomic nervous system activity in pupil size dynamics during sleep⁵³.**

l.691-692: I believe there is a typo in the unit of this sentence:

“...level. This burst was presented every second for a duration of 10 seconds (ON window) followed by a period of silence (OFF window) where no auditory stimulation was presented....”

Changed to: This burst was presented 10 times in one second intervals (ON window) followed by a period of silence (OFF window) where no auditory stimulation was presented.

l.299 ‘N2 that were tones were’, omit ‘that’

Changed to: N2 where tones were

l. 300, the term “general N2 levels” is first used here. Since this description ‘slips in’ as a frequently used description, but is not clearly defined, I suggest to add in brackets here e.g., (i.e., pupil size during N2, excluding the 5 s before and 5 s after an auditory stimulation).

To clearly define the term general N2 levels we now preface the referenced sentence as follows: “... during periods of N2 where tones were not played (general N2 levels). Pupil size was not significantly different to general N2 levels prior to K-complexes”